# Immune phenotypes that are associated with subsequent COVID-19 severity inferred from post-recovery samples

Thomas Liechti [1] ✉, Yaser Iftikhar[1], Massimo Mangino [2,3], Margaret Beddall[1], Charles W. Goss[4], Jane A. O'Halloran [5], Philip A. Mudd [6,7] & Mario Roederer [1] ✉

Severe COVID-19 causes profound immune perturbations, but pre-infection immune signatures contributing to severe COVID-19 remain unknown. Genome-wide association studies (GWAS) identified strong associations between severe disease and several chemokine receptors and molecules from the type I interferon pathway. Here, we define immune signatures associated with severe COVID-19 using high-dimensional flow cytometry. We measure the cells of the peripheral immune system from individuals who recovered from mild, moderate, severe or critical COVID-19 and focused only on those immune signatures returning to steady-state. Individuals that suffered from severe COVID-19 show reduced frequencies of T cell, mucosal-associated invariant T cell (MAIT) and dendritic cell (DC) subsets and altered chemokine receptor expression on several subsets, such as reduced levels of CCR1 and CCR2 on monocyte subsets. Furthermore, we find reduced frequencies of type I interferon-producing plasmacytoid DCs and altered IFNAR2 expression on several myeloid cells in individuals recovered from severe COVID-19. Thus, these data identify potential immune mechanisms contributing to severe COVID-19.

The recent COVID-19 pandemic caused an unprecedented global health crisis. Demographic and socioeconomical factors affect disease severity and mortality[1]. Underlying health conditions such obesity and diabetes or gender with higher risk for males have been associated with disease severity[1]. In addition, genetic predisposition contributes to the development of severe COVID-19[2,3]. GWAS identified several genes encoding for pro-inflammatory chemokine receptors and molecules from the type I interferon pathway, such as OAS1, DPP9, TYK2 and IFNAR2, that associate with the development of severe COVID-19[2,3]. Thus, tissue distribution of immune cells and the responsiveness of innate immunity to infection may be key factors to prevent severe outcome in COVID-19. While GWAS enable the identification of associations between genetic variants and disease severity, such studies fall short of providing insights into the mechanisms by which these genetic traits manifest disease susceptibility. Nearly all of the SNPs identified in GWAS are regulatory and not coding in nature; the altered regulation could be expressed on subsets of immune cells rather than organism-wide. Thus, immunological studies such as immunophenotyping at the single cell level are necessary to gain mechanistic understanding of how genetics affect immune responses[4].

[1]ImmunoTechnology Section, Vaccine Research Center, NIAID, NIH, Maryland 20892, USA. [2]Department of Twin Research & Genetic Epidemiology, King's College of London, London, UK. [3]NIHR Biomedical Research Centre at Guy's and St Thomas' Foundation Trust, London SE1 9RT, UK. [4]Division of Biostatistics, Washington University School of Medicine, St. Louis, MO, USA. [5]Division of Infectious Diseases, Department of Internal Medicine, Washington University School of Medicine, St. Louis, MO, USA. [6]Department of Emergency Medicine, Washington University School of Medicine, St. Louis, MO 63110, USA. [7]Center for Vaccines and Immunity to Microbial Pathogens, Washington University School of Medicine, St. Louis, MO 63110, USA. ✉e-mail: liechti.thom@gmail.com; roederer@nih.gov

Chemokine receptors are crucial in regulating leukocyte trafficking and thereby orchestrating immune responses[5,6]. Thus, chemokine receptors are critical in all aspects of immune responses including adaptive immunity in lymphoid organs[6], early influx of innate immune cells[7] and migration of cells in inflamed tissues[8]. Their expression is tightly regulated and depends on the immune milieu[5]. Imbalance or perturbations in the homeostasis of chemokine and chemokine receptor expression are associated with inflammatory and auto-immune diseases[8]. The pro-inflammatory chemokine receptors CCR1, CCR2, CXCR3 and CX3CR1 are expressed on myeloid cells[9,10], T cells[11,12], and NK cells[13-16]. These chemokine receptors orchestrate immune responses in the lungs including allergies[17], M. tuberculosis[18], and Influenza[16]. In contrast, CCR4[19,20], CCR5[21,22], and CXCR6[23] are mainly expressed on conventional and innate T cells and regulate cell influx to the lung and mucosal tissues[21-23]. CCR4 is critical for homing of immune cells to skin[19,20], while CCR9 regulates cell migration to mucosal tissues[20,24]. Thus, chemokine receptors can induce non-redundant, tissue-specific cell migration. Severe COVID-19 has been associated with perturbations in chemokine levels as well as expression of chemokine receptors highlighting their importance in immunopathology[25,26].

The innate immune system ensures rapid and effective immune responses against viruses and is impaired in severe COVID-19[27-29]. IFNAR2 is critical for type I interferon mediated immunity; homozygous mutations, which abrogate IFNAR2 expression, are associated with fatal outcome in viral infections[30]. The activity of type 1 interferon remains controversial in SARS-CoV2[31]. Severe COVID-19 is associated with low serum levels of type I interferon[32]. In contrast, robust type I interferon response occurs in lung tissues from severe but not mild COVID-19 cases[33]. Furthermore, neutralizing autoantibodies against type I interferon[34] or loss-of-function mutations in type I interferon pathway[35] occur more frequently within severe COVID-19 cases. Thus, while excessive type I interferon response may exacerbate inflammation and severity of COVID-19, it is likely that the lack thereof is also detrimental.

Based on the GWAS data[2,3] we hypothesized that immune signatures at steady-state (i.e., prior to infection and following recovery) impact the outcome of COVID-19 severity. This may manifest as a variety of phenotypes: altered level of expression or altered regulation of certain subsets of immune cells. Here we test this hypothesis using high-dimensional, comprehensive immunophenotyping in peripheral blood mononuclear cells (PBMC) of individuals that recovered or substantially improved from mild, moderate, severe and critical COVID-19 (Supplementary Fig. 1 and Supplementary Table 1). Particularly, we focus on the expression of chemokine receptors and IFNAR2 identified by GWAS[2,3]. We identify several immune signatures at steady-state which differ between individuals recovered from non-severe and severe COVID-19. This included altered expression of various chemokine receptors on NK and MAIT cells as well as altered abundance of innate immune subsets. In addition, our data show reduced levels of plasmacytoid DCs (pDC), which produce high levels of type I interferon[36] and increased expression of IFNAR2 on several myeloid cell subsets at steady-state in individuals recovered from severe COVID-19, pointing towards impaired type I interferon responsiveness. Thus, these data define predictable immune signatures associated with severe COVID-19 outcome and improve our understanding of pathogenesis of COVID-19.

## Results

### Expression profile of chemokine receptors, IFNAR2 and functional receptors

We assessed the immune profile in PBMC from 173 unexposed healthy individuals (referred to as healthy controls) using 28-color flow cytometry (Fig. 1a and Supplementary Tables 2, 3). We measured immune cell subsets with two backbone panels focusing on either

B cells and myeloid cells or innate-like and conventional T cells as well as NK cells (referred to as BDC and TNK panels, respectively; Supplementary Table 2). We used each backbone panel with two sets of chemokine receptors (CR1 and CR2). Thus, we measured 4 sets of distinct markers for each sample (Supplementary Tables 2, 3). Manual definition of immune subsets and functional marker expression profile on these subsets are shown in supplementary material (Supplementary Figs. 2–7).

Immune subsets showed heterogenous expression of various chemokine receptors, the Ecto-NTPDase CD39, co-stimulatory receptors CD40 and CD86, Interferon-alpha receptor 2 (IFNAR2) and co-inhibitory molecule TIGIT (Fig. 1a). We focused our subsequent analysis on immune traits for which the lineage showed discernible expression. For instance, XCR1 and CCR3 were only expressed on cDC1s and Basophils, respectively, while B cells did not express CCR1, CCR2, CCR3, CCR4, CCR8, CXCR6, and CX3CR1. The remaining 1758 out of 3787 immune traits consisted of frequency of immune cell subsets ($N = 349$), cells expressing functional markers ($N = 620$) or the mean fluorescence intensity (MFI; $N = 789$) of functional markers.

### Prolonged immune perturbations after recovery from COVID-19

We aimed to identify immune signatures at steady-state which contribute to severe COVID-19. However, cohorts with baseline PBMC samples from patients who had not yet been infected with COVID-19 are not available. In this cross-sectional study, we analyzed PBMC collected after recovery from mild, moderate, severe and critical COVID-19 (Supplementary Fig. 1 and Supplementary Table 1) and focused on traits related to the highly significant results from GWAS analysis.

COVID-19 induced immune perturbations can persist after viral clearance and recovery[37,38]. We hypothesized that immune cells may show three potential trajectories following infection. First, a cell trait may not be affected by COVID-19 infection and remain at baseline throughout infection (but still contribute to susceptibility to severe disease). Secondly, a trait may be affected and deviate from unexposed healthy individuals only during active COVID-19 and recover over time after pathogen clearance. Lastly, a trait may change rapidly after viral transmission and remain perturbed after viral clearance without improvement. The latter is likely infrequent, if at all, due to the renewal capacity of the human immune system[39-44]. The second scenario would result in temporal changes of immune traits which would be detectable in samples collected early vs late after symptom onset. To identify traits that might contribute to COVID-19 severity, we took the conservative approach of eliminating traits that showed any evidence of perturbation over time, for which we cannot assert the baseline, pre-COVID-19 value. A caveat to this approach is that a trait which changed from baseline before our earliest measurements, and then remained at that level following our latest measurements, would be included in our analyses. However, we considered that scenario highly unlikely in this dynamic acute infection, particular since viral clearance often occurs early irrespective of disease severity[37] and given the immune system's capacity to recover rapidly from perturbations[39-44].

We first identified persistent immune perturbations which may contribute to long-lasting COVID19-related symptoms known as long COVID[37,38]. We focused on the moderate and severe COVID-19 group as these groups showed the largest time range between symptom onset and sample collection (Supplementary Fig. 1a; Moderate, 24–129 days; Severe, 16–184 days). As shown, our analysis enables the identification of temporal changes of immune traits over a period of ~120–180 days post infection. We applied two strategies to identify persistently affected immune traits. These included (i) linear regression of immune traits and time between symptom onset and sample collection, and (ii) comparison of samples collected before and after 60 days of symptom onset using a Wilcoxon test. The two strategies showed similar results (Fig. 1b). We assessed the top hits from both analyses ($p < 0.001$ in at

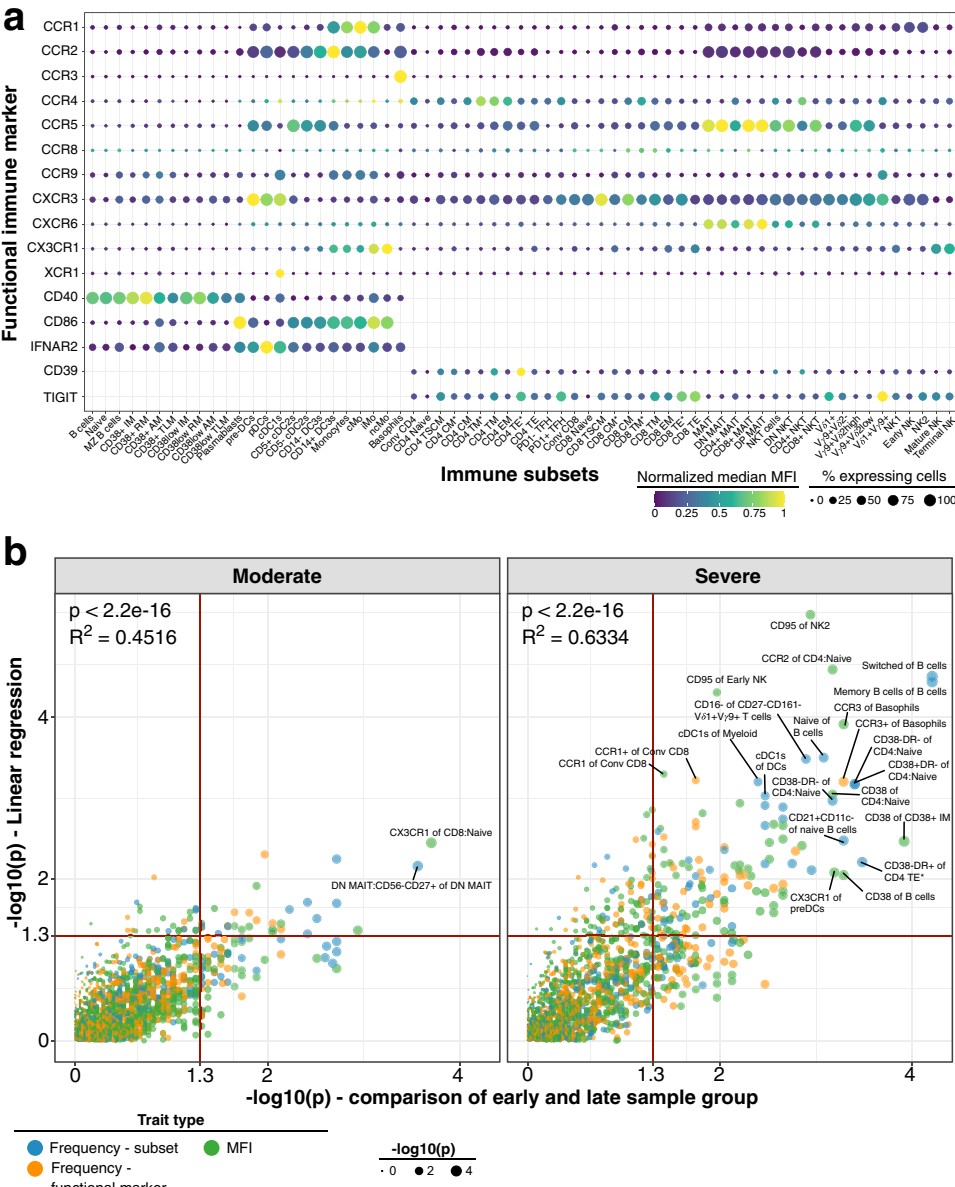

**Fig. 1 | Expression of functional markers and temporal dynamics of immune traits in COVID-19. a** Expression of chemokine receptors, CD40, CD86, IFNAR2, CD39 and TIGIT (rows) on immune cell populations (columns) is depicted. Median of mean fluorescence intensities (MFI) derived from 173 unexposed healthy individuals is visualized by min-max normalized color gradient. Dot size corresponds to median percentage of cells expressing these markers. Missing dots indicate that marker was not measured. **b** Immune traits (*N* = 1758) at baseline or affected by long-term perturbations were distinguished in individuals recovered from moderate (left) and severe (right) COVID-19 cases. A combination of (I) linear regression

analysis between immune traits and days between symptom onset and sample collection and (II) comparison of samples collected before and after 60 days of symptom onset using a two-sided Wilcoxon test was used as described in Online methods. Plot shows unadjusted −log10 *P*-values from both analyses. Dot size increases with significance from Wilcoxon test. Trait types are colored (Frequency of immune subset in blue, Frequency of expressing functional marker in orange and MFI values in green). Red line highlights threshold for unadjusted significance (*P* = 0.05). Source data are provided as a Source Data file.

least one analysis, *N* = 24) to further delineate persistent immune perturbations in COVID-19 (Fig. 2a).

The most prominent persisting perturbations occurred within switched (containing memory B cells and plasmablasts) and memory CD20$^+$IgD$^-$CD38$^{-/+}$CD27$^{-/+}$ B cells (Fig. 2a). Switched and naive B cells did not change in moderate COVID-19 over time but significantly decreased and increased, respectively, in severe cases (Spearman's rank correlation; Naïve: $R^2$ = 0.36, *P* = 0.002; Switched: $R^2$ = 0.44, *P* = 4 × 10$^{-4}$) to levels observed in unexposed healthy individuals (Fig. 2b). Both naïve and switched B cells did not differ between study groups (Fig. 2b). Similar dynamics occurred for CD38$^+$HLA-DR$^-$ and CD38$^-$HLA-DR$^-$ CD4 naïve T cells which showed an increase and

decrease, respectively, over time in the severe COVID-19 group (Spearman's rank correlation; CD38$^+$HLA-DR$^-$ CD4 naive: $R^2$ = 0.42, *P* = 5.8 × 10$^{-4}$; CD38$^-$HLA-DR$^-$ CD4 naive: $R^2$ = 0.42, *P* < 6.5 × 10$^{-4}$) with later timepoints reaching levels observed in unexposed healthy individuals (Fig. 2c). In addition, decreased CD38$^+$HLA-DR$^-$ and increased CD38$^-$HLA-DR$^-$ CD4 naïve T cells occurred in individuals recovered from severe and critical COVID-19 (Bonferroni-adjusted *P*-value range 0.02–1.46 × 10$^{-4}$) (Fig. 2c). CD38 on naive T cells is downregulated in HIV-1 but their functional properties remain unclear[45,46].

Cross-presenting cDC1s induce potent CD8 T cell responses[47]. Timepoints early after onset of symptoms had reduced levels of cDC1s in severe COVID-19 cases, but these increased later to levels observed

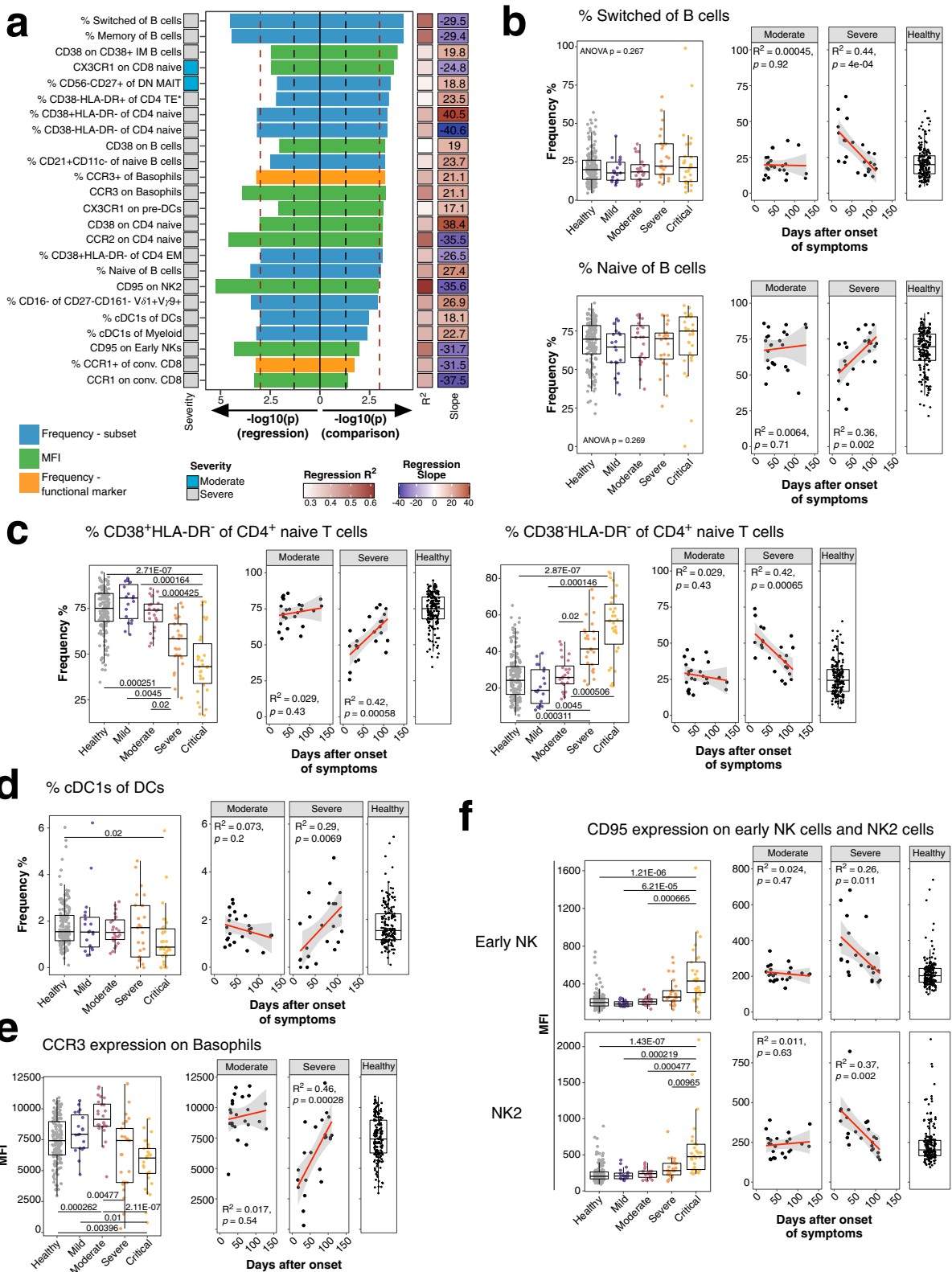

in unexposed healthy individuals, suggesting perturbations of cDC1s during active COVID-19 (Spearman's rank correlation; $R^2 = 0.29$, $P = 0.0069$) (Fig. 2d). We also observed changes in the expression levels of receptors over time (Fig. 2a, e, f). Basophils regulate $T_H2$ immune responses[48] and express CCR3 which induces cell migration and release of histamine and leukotriene[49]. Basophils expressed reduced levels of CCR3 early after symptom onset while levels were

closer to unexposed healthy individuals at later timepoints within the severe COVID-19 group (Spearman's rank correlation; $R^2 = 0.46$, $P = 2.8 \times 10^{-4}$) (Fig. 2e). CCR3 expression was reduced in basophils from severe and critical COVID-19 cases (Bonferroni-adjusted $P$-value range $0.01 – 2.11 \times 10^{-7}$). Furthermore, CD95 expression decreased over time in early NK (Spearman's rank correlation; $R^2 = 0.26$, $P = 0.011$) and NK2 cells (Spearman's rank correlation; $R^2 = 0.37$, $P = 0.002$) in severe

**Fig. 2 | Long-term perturbations of immune traits in COVID-19. a** Top immune traits affected by long-term perturbations are depicted. Traits are derived from analysis in Fig. 1b and selected for *P*-value < 0.001 in one of both analysis (linear regression and/or Wilcoxon test). Bars pointing to the left and right are derived from linear regression and Wilcoxon test, respectively, and are colored based on trait type (Frequency of immune subset in blue, Frequency of expressing functional marker in orange and MFI values in green). Colored bar on the left depicts severity group from which the significant trait is derived. $R^2$ and slope from linear regression are shown as colored bars on the right. Values in the right bar are slope values from linear regression. Red and black dashed lines show *P*-value cut-off of 0.001 and 0.05, respectively. **b** Frequencies of switched (top row) and naïve (bottom row) B cells of total B cells are shown. Plot on the left shows frequencies as boxplots for unexposed healthy donors (gray) and individuals recovered from mild (purple), moderate (burgundy), severe (orange) or critical COVID-19 (yellow). The two plots on the right show the frequency of cells as a function of time between symptom onset and sample collection for individuals recovered from moderate and severe COVID-19. Far right plot shows the distribution of the traits in 173 unexposed healthy individuals. Similar to Fig. 2b, dynamics of **c** CD38⁺HLA-DR⁻ (left) and CD38⁻HLA-DR⁻ of CD4 naive T cells (right), **d** frequencies of cDC1s of total DCs, **e** CCR3 MFI of basophils and **f** CD95 MFI of early NK and NK2 cells are shown. Age-corrected residuals from linear regression were used for statistical analysis. For comparison between groups, one-way ANOVA was used on residuals to test for overall significant difference prior to two-sided Wilcoxon test with Bonferroni correction. Boxplots depict median and interquartile range (IQR) and length of whiskers is 1.5 times IQR. Second and third plot show dot plots with linear regression (red line) and 95% confidence interval for individuals recovered from moderate and severe COVID-19, respectively. Source data are provided as a Source Data file.

COVID-19 (Fig. 2f). CD95 expression was significantly elevated in both subsets from critical COVID-19 compared to all other groups (Bonferroni-adjusted *P*-value range 0.00965– $1.43 \times 10^{-7}$). Fas (CD95) induces cell death and is important for immune homeostasis[50] and is expressed on memory and effector T cells[45]. In conclusion several immune traits in severe COVID-19 required prolonged time—up to 100 days after symptom onset—to reach baseline levels which can be several months which agrees with previous studies[37,38].

## Predictive potential of lymphocyte immune traits

Next, we hypothesized that stable immune traits ($N = 1365$) between symptom onset and sample collection remained at or returned early to pre-infection baseline. We defined stable immune traits as traits which did not show any significant temporal changes in either the moderate or severe COVID-19 group and either the linear regression analysis or the comparison of early and late group (Fig. 1b). As described above, our analysis is conservative in identifying immune traits which change over time. We aimed to identify differences in stable traits between individuals recovered from mild ($N = 19$) and moderate ($N = 24$) COVID-19 (combined and referred to as non-severe group, $N = 43$) and severe ($N = 25$) and critical ($N = 30$) COVID-19 cases (combined and referred to as severe group, $N = 55$). Such differences may give clues about pre-infection immune signatures which favor the development of severe COVID-19. We identified distinctive immune features between these two groups using logistic regression ($N = 150$, FDR-adjusted *P*-value cut-off <0.01) as described in the Online methods (Supplementary Fig. 8). All significant results from this analysis are listed in Supplementary Fig. 8b. Despite substantial improvement, some patients from the severe ($N = 6$) and critical ($N = 21$) COVID-19 group were still hospitalized at sample collection (Supplementary Fig. 1d, e). These samples may bias the analysis due to persistent immune perturbations or pathologies; we therefore repeated the analysis and only included individuals which were discharged prior to or at the day of sample collection (Supplementary Fig. 9a, b). All significant results from this analysis are listed in Supplementary Fig. 9b. We obtained similar results with this smaller sample set (Non-severe, $N = 43$; Severe, $N = 28$, FDR-adjusted *P*-value cut-off <0.012), compared to all individuals, with highly correlated *P*-values between both analyses (Spearman's rank correlation, $R = 0.9$, $P < 2 \times 10^{-16}$) (Supplementary Fig. 9c). In fact, 65 significant hits (FDR-adjusted $P_{ALL} < 0.01$, FDR-adjusted $P_{Non-hospitalized} < 0.012$) were shared between these two analyses using either all patients or only non-hospitalized patients at the time of sample collection (Supplementary Fig. 9d). Non-hospitalized refers to individuals who did not require hospitalization or were discharged from the hospital prior to or at day of sample collection. Information about hospitalization can be found in Supplementary Fig. 1e and Supplementary Table 1.

Only 6 further immune traits were discovered with the non-hospitalized sample set (FDR-adjusted $P < 0.012$). However, 85 significant immune traits (FDR-adjusted $P < 0.0017$) were only discovered when all patients were analyzed. This may be due to the lower statistical power with the smaller sample set as suggested by the strong correlation of *P*-values (Supplementary Fig. 9c).

We primarily focused our analysis on traits which significantly differed between non-severe and severe COVID-19 cases in both sample sets (all vs. non-hospitalized, $N = 65$, FDR-adjusted $P_{ALL} < 0.01$, FDR-adjusted $P_{Non-hospitalized} < 0.012$) (Supplementary Figs. 8b, 9b). NK cells are critical for antiviral defense[51] and impaired in severe COVID-19[52]. We discovered several chemokine receptor signatures on NK cells ($N = 8$) associated with the development of severe COVID-19, including upregulated CX3CR1 expression on early NK cells (Fig. 3a) and increased levels of CCR4, CCR9 and CXCR3 on terminal NK cells (Fig. 3b). However, the expression of these molecules by other cell types were not associated with severity, underscoring the need to perform multiparameter analysis at the single cell level.

We also identified several potentially predictive traits ($N = 75$) within conventional T cells. Naïve and transitional memory (TM) CD8⁺ T cells from individuals suffered from severe and critical COVID-19 expressed higher levels of CCR4 (Bonferroni-adjusted *P*-value range $0.03–7 \times 10^{-5}$). This pattern did not occur on naive and TM CD4⁺ T cells (Fig. 3c). The co-inhibitory receptor TIGIT limits T cell activation and viral immunopathology[53]. Stem cell-like memory (TSCM), central memory (CM) and terminal effector* (TE*) CD8⁺ T cells exhibited reduced TIGIT expression in individuals recovered from severe and critical COVID-19 (Bonferroni-adjusted *P*-value range $0.01–7.21 \times 10^{-6}$) (Fig. 3d). In contrast, naïve CD8⁺ T cells and MAIT cells expressed similar levels between study groups or elevated levels of TIGIT in individuals suffered from severe and critical COVID-19, respectively (Bonferroni-adjusted *P*-value range 0.04–0.004).

MAIT cells recognize unique microbial riboflavin-like antigens[54], respond rapidly through the expression of cytokines and cytotoxic granzyme B[55] and have various functions including pathogen clearance, tissue repair but also immunopathology and inflammation[54]. The frequency of MAIT cells was decreased in severe and critical COVID-19 (Supplementary Fig. 9e) (Bonferroni-adjusted *P*-value range $3.99 \times 10^{-4}–2.96 \times 10^{-6}$). Furthermore, more individuals recovered from severe and critical COVID-19 showed reduced frequencies of central memory (CM) CD4⁺ and CD8⁺ T cells (defined as CD45RA⁻CCR7⁺CD27⁺) (Bonferroni-adjusted *P*-value range $0.02–7.61 \times 10^{-6}$) (Supplementary Fig. 9f). In addition, individuals recovered from critical COVID-19 had elevated levels of activated (defined as CD38⁺HLA-DR⁺) CD8⁺ effector and terminal memory T cells (Bonferroni-adjusted *P*-value range $5.46 \times 10^{-4}–1.04 \times 10^{-6}$) (Fig. 3e).

We did not identify many B cell traits predictive for COVID-19 severity. However, patients with severe COVID-19 had lower baseline frequencies of marginal zone (MZ) B cells, which produce natural IgM mostly targeting bacterial glycans and are considered an early wave of immune defense (Fig. 3f) (Bonferroni-adjusted *P*-value range $0.00192–1.84 \times 10^{-4}$)[56].

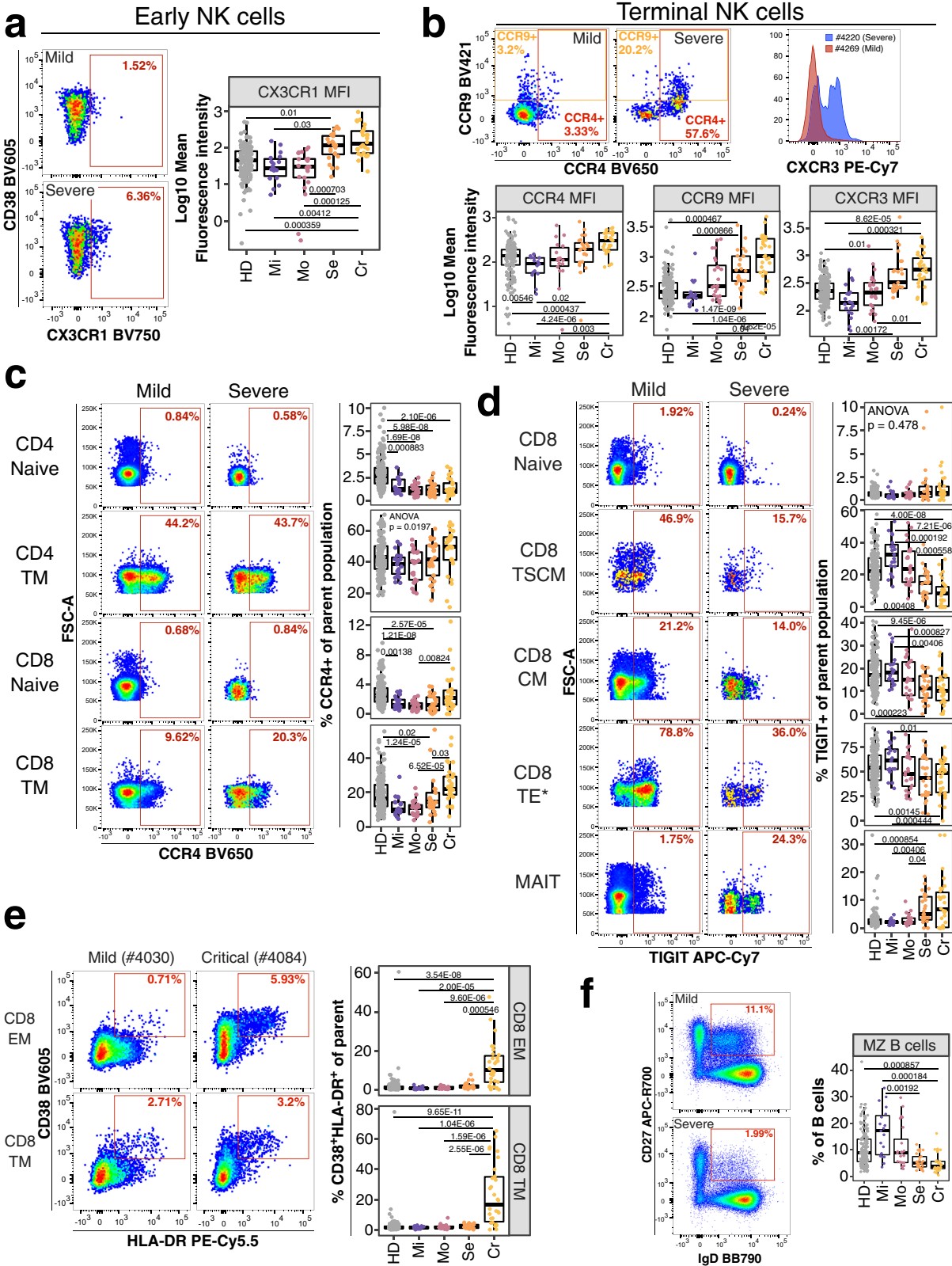

## Predictive potential of myeloid immune traits

Innate immune signatures determine the trajectories of disease severity early during active COVID-19[57]. We assessed several innate immune subsets such as monocytes and dendritic cells (DCs) in the periphery[9,58] as well as several critical markers for stimulation of adaptive immune responses including CD40 and CD86[59,60]. Individuals recovered from severe and critical COVID-19 had reduced frequencies of plasmacytoid DCs (pDCs) and CD14+ DC3s (Bonferroni-adjusted P-value range 0.00226–3.71 × 10⁻⁷) (Fig. 4a). CD14⁻ and CD14⁺ DC3s are inflammatory cells and more efficient in activating and polarizing T helper responses compared to conventional cDC2s[9].

The chemokine receptor profile on dendritic cells did not differ substantially between individuals recovered from non-severe and severe COVID-19. We observed increased expression of CX3CR1 on

**Fig. 3 | Potential immune features at baseline predicting COVID-19 severity.**
**a** Flow cytometry data (left) depicts CX3CR1 expression on early NK cells from an individual recovered from mild (top) and severe (bottom) COVID-19. Quantification of mean fluorescence intensity (MFI) of CX3CR1 on early NK cells is shown as boxplot for all study groups. **b** Flow cytometry dot plots on top row (left and middle plot) depicts expression of CCR4 and CCR9 for a mild and severe COVID-19 case. Histogram overlay shows CXCR3 expression for the same cell subset and donors. MFI values for the same receptors are shown as boxplots for all groups (bottom row). **c** Flow cytometry plot depicts expression of CCR4 on CD4$^+$ naive (top row), CD4$^+$ transitional memory (TM, second row), CD8$^+$ naïve (third row) and CD8$^+$ TM (bottom row) T cells from an individual recovered from mild (left column) or severe (right column) COVID-19. Quantification of these subsets in all study groups are shown as boxplots (right). **d** Flow cytometry plot depicts TIGIT expression on CD8$^+$ naïve (top row), CD8$^+$ stem-cell like memory (TSCM, second row), CD8$^+$ central memory (CM, third row), CD8$^+$ terminal effector* (TE*, fourth row) T cells and MAIT cells (bottom row) from an individual recovered from mild (left column) and

severe (right column) COVID-19. Quantification of these subsets in all study groups are shown as boxplots (right). **e** Flow cytometry data (left) depicts CD38 and HLA-DR expression on CD8 effector (EM; top) and terminal (TM; bottom) memory T cells from an individual recovered from mild (left) and critical (right) COVID-19. The gate defines CD38$^+$HLA-DR$^+$ activated T cells. Quantification of these subsets in all study groups are shown as boxplots (right). **f** Flow cytometry example data (left) for gating of marginal zone (MZ) B cells from total B cells in an individual recovered from mild and severe COVID-19 is shown. Boxplot (right) shows frequencies of MZ B cells of total B cells in all study groups. Residuals from linear regression between immune trait and age were used to calculate statistics on age-corrected data. ANOVA with subsequent two-sided Wilcoxon test and Bonferroni correction on residuals was performed for statistics highlighted in boxplots. Boxplots depict median and interquartile range (IQR) and length of whiskers is 1.5 times IQR. Study groups encompass healthy unexposed controls (HD) and individuals recovered from mild (Mi), moderate (Mo), severe (Se) and critical (Cr) COVID-19. Source data are provided as a Source Data file.

pDCs and cross-presenting cDC1s associated with disease severity (Bonferroni-adjusted P-value range $0.00301–1.83 \times 10^{-5}$) (Fig. 4b). Frequency of monocyte subsets did not differ between groups. However, classical and intermediate monocytes from individuals recovered from severe COVID-19 had reduced expression of pro-inflammatory chemokine receptors CCR1 and CCR2 (Bonferroni-adjusted P-value range $0.03–2.07 \times 10^{-9}$) (Fig. 4c). In contrast, non-classical pro-inflammatory monocytes showed no differences of CCR1 and CCR2 expression between COVID-19 severity groups (Fig. 4c).

Genome-wide association studies identified IFNAR2 as a risk factor for severe COVID-19[2,3]. Furthermore, type I interferon response is critical for effective immune responses against COVID-19[28,29,31,32]. We measured expression of IFNAR2 on monocytes, dendritic cells and B cells. IFNAR2 expression was lowest on naïve B cells and highest on pDCs and cDC1s within the 173 unexposed healthy individuals (Fig. 1a). Noteworthy, IFNAR2 expression could be detected on most subsets including cDC2s, DC3s, and monocyte subsets (Fig. 1a). Next, we assessed how the expression of IFNAR2 on these immune cell subsets differed between the unexposed healthy controls and the different COVID-19 severity groups (Fig. 4d, e). Monocytes and dendritic cells, except cDC1s and pDCs, showed elevated expression of IFNAR2 in individuals recovered from severe and critical COVID-19 compared to unexposed healthy controls and mild and moderate COVID-19 group (Bonferroni-adjusted P-value range $0.04–4.95 \times 10^{-5}$) (Fig. 4d, e).

In non-myeloid cells, IFNAR2 expression was elevated in basophils but no substantial change in expression of IFNAR2 occurred in other non-myeloid cells with disease severity (Fig. 4d, e). IFNAR2 was slightly reduced in several CD38$^{low}$ memory B cell populations severe and critical COVID-19 (Fig. 4d). However, these CD38$^{low}$ memory B cell subsets were not significantly different between non-severe and severe COVID-19 group (Supplementary Fig. 8b).

### Unsupervised cluster analysis

Next, we used unsupervised clustering to extend our analysis and identify potential immune signatures not revealed by our manual gating analysis. We split cells from both chemokine receptor panels into main lineages based on manual gating and defined 388 clusters using FlowSOM as described in the Online methods (Supplementary Figs. 10–12 and Supplementary Tables 4, 5). Subsequently, we excluded persistently perturbed immune clusters ($N = 97$) as described for manually defined traits in Fig. 1b (Supplementary Fig. 13). Next, we identified distinct immune traits between non-severe and severe COVID-19 after recovery using logistic regression (Fig. 5a, b). Results with all samples and with only the non-hospitalized individuals strongly correlated (Spearman's rank correlation, $R = 0.9$, P < 2.2e−16) confirming that hospitalization was not a major driver (Supplementary Fig. 13b, c).

We focused on 42 significant clusters ($p_{FDR} < 0.01$) across all lineages (Fig. 5a, b). From each chemokine receptor panel (CR1 and CR2) 5 and 6 significant clusters ($p_{FDR} < 0.01$) resembled innate-like T cells, respectively. Four clusters (clusters 34, 35, 37, and 38) from CR1 and one (cluster 3) from CR2 panel were MAIT cells as defined by T cell receptor (TCR) V$\alpha$7.2 and CD161 (Fig. 5c). These clusters were reduced in severe COVID-19 (Bonferroni-adjusted P-value range $0.04–3.71 \times 10^{-7}$) (Fig. 5d) matching the overall decreased frequency of MAIT cells (Supplementary Fig. 9e). V$\delta$2V$\gamma$9T cell clusters 20 and 25 from CR2 panel were expanded in severe and critical COVID-19 (Bonferroni-adjusted P-value range $0.00564–6.39 \times 10^{-8}$) and characterized by CCR9, CXCR3 and TIGIT expression (Fig. 5d and Supplementary Fig. 14). In contrast V$\delta$2 V$\gamma$9 T cell cluster 24 expressed higher levels of CCR4 and CCR8 but lacked CXCR3 and TIGIT (Fig. 5d and Supplementary Fig. 14). V$\gamma$9V$\delta$2 T cells recognize unique microbial phospho-antigens and respond rapidly through secretion of cytokines or cytotoxic molecules and thus may exacerbate inflammation[61]. Noteworthy, we observed a shift from CCR7$^+$CD27$^+$ naïve-like V$\delta$2V$\gamma$9 T cells (CR2 cluster 30) towards effector-like cells (CR2 clusters 20, 24, and 25) which lack CCR7 and CD27 in individuals recovered from severe and critical COVID-19 (Fig. 5d)[62].

Within myeloid cells, CD123$^+$CD5$^-$ pDCs (CR1: 24, CR2: 28) and CD123$^+$CD5$^+$ pre-DCs (CR1: 28, CR2: 29) were significantly reduced (Bonferroni-adjusted P-value range $0.04–4.11 \times 10^{-6}$) in individuals recovered from severe and critical COVID-19 (Fig. 6a, b). The latter are precursors and differentiate into conventional cDC1 and cDC2[9]. Both cell subsets were characterized by expression of CD38, CCR5 and high levels of CXCR3 (Fig. 6b and Supplementary Fig. 15). CCR1, CCR2 and IFNAR2 were expressed at higher levels on pDCs while co-stimulatory CD86 was lower and CD40 expression was lacking on both (Fig. 6b and Supplementary Fig. 15). CD14$^-$ DC3s (CR1: 22; CR2: 21) and CD14$^+$ DC3s (CR1: 11) differ between individuals recovered from non-severe and severe COVID-19 (Fig. 6a, b) in agreement with our manual analysis (Fig. 4a and Supplementary Fig. 8).

We further examined myeloid cells from mild and severe COVID-19 cases using tSNE. Clusters shown in Fig. 6a exhibited reduced density on the tSNE map in severe cases (Fig. 6c and Supplementary Fig. 10a–c). We analyzed the relationship between the subsets identified in panels CR1 and CR2 which showed high overlap suggesting the identification of similar populations with both panels (Fig. 6d).

Overall, unsupervised analysis indicates similar immune subsets which differ between individuals recovered from non-severe and severe COVID-19 compared to the manually defined subsets. However, the unsupervised analysis enabled more detailed insights into the unique combinatorial expression patterns of chemokine receptors and other functional molecules on these subsets.

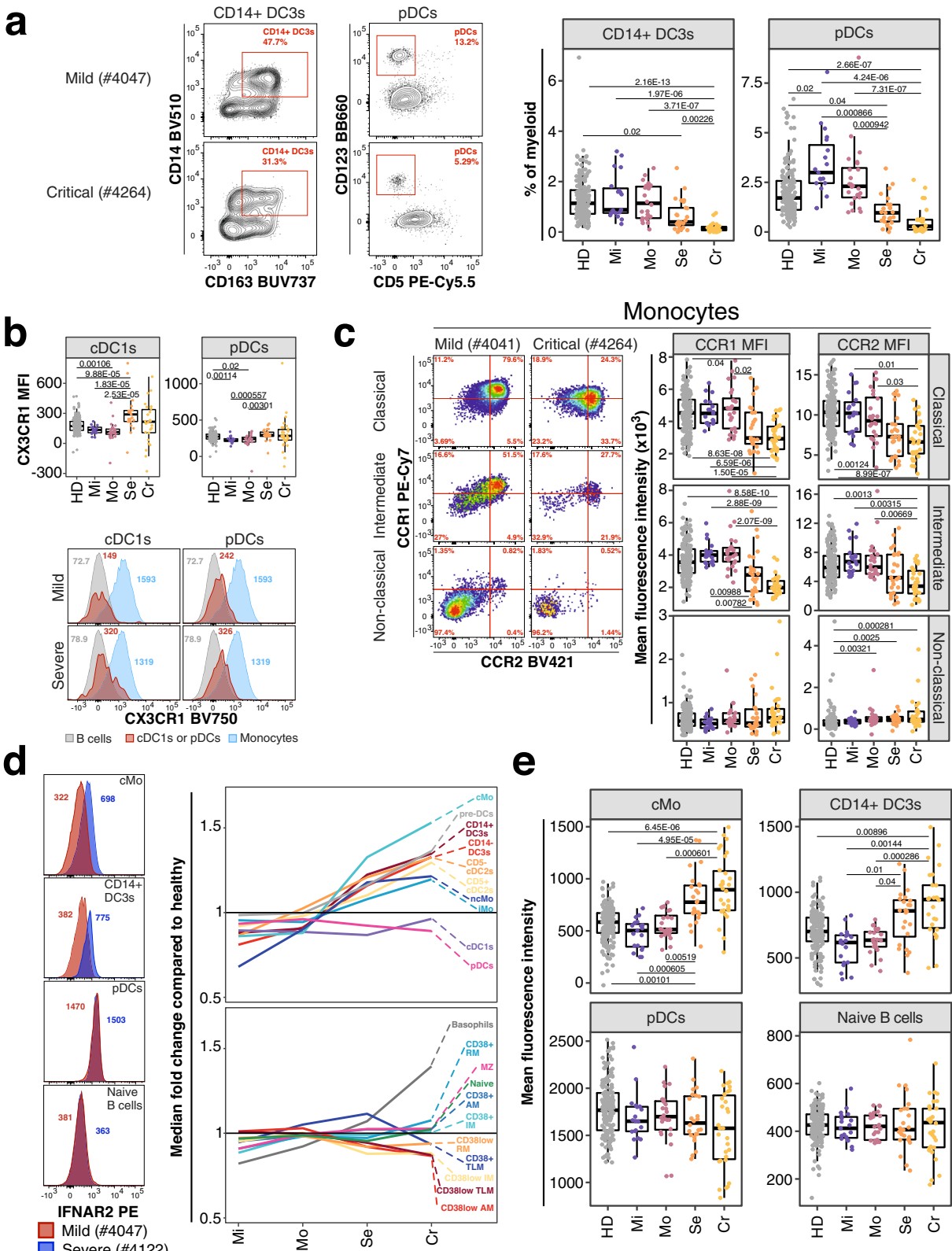

## Discussion

We lack mechanistic insights into how pre-infection immune signatures contribute to the development of life-threatening COVID-19. GWAS identified several genes associated with COVID-19 severity[2,3]. The most predictive genes encode for pro-inflammatory chemokines such as CCR2, CCR3, CXCR6 and XCR1 and molecules from the type I interferon pathway including

IFNAR2[2,3]. However, these GWAS associations do not indicate potential mechanisms (e.g., altered expression of CCRs on subsets of leukocytes). Thus, immunological studies such as immunophenotyping are needed to better understand the mechanisms by which these immune traits impact disease severity. In addition, most studies focused on finding distinctive immune signatures during active severe COVID-19[32,33,52,57,63,64]. Here, we hypothesized

**Fig. 4 | Innate immune signatures predict COVID-19 severity. a** Example flow cytometry data for frequencies of pDCs from dendritic cells and inflammatory CD14[+] DC3s of total DC3s is shown from an individual recovered from mild (top) and critical (bottom) COVID-19. Corresponding enumeration for all donors based on study group and as percentage of total myeloid cells are shown as boxplots (right). Precise delineation of pDCs is shown in Supplementary Fig. 1c. **b** Mean fluorescence intensity (MFI) of CX3CR1 on cDC1s (left) and pDCs (right) is shown for all study groups as boxplot (top row). Example flow cytometry data for CX3CR1 signal (red peak) on cDC1s (first column) and pDCs (second column) is shown as histogram for an individual recovered from mild (top row) and severe (bottom row) COVID-19. B cells (gray) and Monocytes (blue) are overlaid as reference populations known to lack and express CX3CR1, respectively. Numbers in histogram plots highlight MFI. **c** Flow cytometry data (left) depicts CCR1 and CCR2 expression on classical (top), intermediate (middle) and non-classical (bottom) monocytes from a patient recovered from mild (left column) and critical (right column) COVID-19. Boxplots (right) show MFI values of CCR1 (first column) and CCR2 (second column) on the same monocyte populations for all study groups. **d** Expression of IFNAR2 from an individual recovered from mild (red) and severe (blue) COVID-19 is shown as overlaid histogram (left) for classical monocytes (top), CD14[+] DC3s (second row), pDCs (third row) and naïve B cells (bottom). Plot on the right depicts fold change of median IFNAR2 expression of each disease severity group compared to median IFNAR2 expression of unexposed healthy individuals on all defined myeloid (top) and non-myeloid (bottom) subsets. **e** Boxplots show IFNAR2 MFI for all study groups for classical monocytes, CD14 + DC3s, pDCs and naïve B cells. Residuals from linear regression between immune trait and age were used to calculate statistics on age-corrected data. ANOVA with subsequent two-sided Wilcoxon test and Bonferroni correction on residuals was performed for statistics highlighted in boxplots. Boxplots depict median and interquartile range (IQR) and length of whiskers is 1.5 times IQR. Study groups encompass healthy unexposed controls (HD) and individuals recovered from mild (Mi), moderate (Mo), severe (Se) and critical (Cr) COVID-19. Source data are provided as a Source Data file.

that pre-infection immune signatures determine the trajectories of COVID-19 severity.

Samples collected prior to infection are most suitable to study pre-infection immune signatures. However, pre-infection samples are difficult to obtain for acute emerging diseases such as COVID-19. Thus, we measured the immune composition using high-dimensional flow cytometry in peripheral blood of individuals recovered from mild, moderate, severe and critical COVID-19 to identify immune signatures associated with COVID-19 severity. The human immune system rapidly reverts to baseline after pathogen clearance with a composition comparable to pre-infection[65,66]. Therefore, probing the immune system after disease recovery serves as an alternative to identify baseline immune signatures which determine disease severity.

Our approach has the caveat that rapidly and permanently perturbed immune signatures will be identified as baseline. However, rapid immune recovery has been described for several acute viral infections including Ebola[67], Zika[68], Dengue[69,70], Measles[71], Hantavirus[72], and Respiratory Syncytial Virus[73] as well as live virus vaccines for Smallpox and Yellow Fever[74]. In these cases, most immune perturbations recovered within weeks after pathogen clearance. Therefore, it is unlikely that immune perturbations occur permanently without any sign of improvement over time. Such traits would, in any case, represent a small fraction of the traits we observed.

Similarly, most SARS-CoV2-induced immune perturbations normalized within 60–80 days after symptom onset[37,38,75]. Furthermore, immune perturbations in nasal tissues from severe COVID-19 patients improved already within 15–61 days of hospitalization and normalized after discharge (median of 77 days after symptom onset). Bergamaschi et al. demonstrated that viral titers drop to undetected levels irrespective of disease severity between 13–24 days after symptom onset which overlapped with improvement of immune perturbations[37]. Nevertheless, some immune perturbations lasted for months and included naïve B cells, activated CD38[+]HLA-DR[+] T cells and the expression of CXCR3[37,38,75]. Noteworthy, not all the long-term immune perturbations described by Bergamaschi et al. could be replicated in studies with a more detailed focus on individual immune cell subsets including pDCs[76] and MAIT cells[77]. In agreement with these studies, we also did not observe any temporal changes of pDC and MAIT cell abundance in individuals recovered from moderate and severe COVID-19. It remains to be determined why there are discrepancies between studies regarding long-term immune perturbations, but reasons could include timing of sample collection and sample cohorts. Overall, based on these studies and our results (Fig. 1b), we argue that samples taken after recovery from COVID-19 mostly reflect the immune system at steady-state and are comparable to pre-infection.

Our study identifies several distinct chemokine receptor signatures between individuals recovered from non-severe (mild/moderate) and severe (severe/critical) COVID-19. Chemokine receptors are important for protective immune responses against viral infections such as West Nile Virus and Influenza[21,78] and their expression is altered in severe COVID-19[57,63]. In our study, individuals recovered from severe COVID-19 had increased expression of lung-homing chemokine receptors CCR4, CXCR3 and CX3CR1 on NK cell subsets (Fig. 3a, b). These receptors result in exacerbated lung inflammation and impaired immune responses against viruses[13,17,79]. In addition, NK cells can facilitate inflammation during viral infections[80]. Thus, increased baseline expression of lung-homing chemokine receptors on NK cells may facilitate NK cell migration and exacerbate lung inflammation in COVID-19. We also identified elevated CCR4 levels on transitional memory CD4[+] and CD8[+] T cells in these individuals (Fig. 3c) highlighting that enhanced homing of T cells to the lung might exacerbate COVID-19.

In contrast, individuals recovered from mild and moderate COVID-19 expressed higher levels of TIGIT (Fig. 3d). TIGIT expression prevents immune pathologies of viral infections in mice and reduces lung damage in influenza infection[53]. Thus, increased levels of TIGIT might have a protective function against severe lung damage and consequently the development of life-threatening COVID-19.

Furthermore, we observed reduced expression of CCR1 and CCR2 on monocyte subsets from individuals recovered from severe COVID-19 (Fig. 4c). CCR2 can have a protective activity in the early phase of mouse-adapted SARS-CoV2 infection[81]. Similarly, CCR1 and CCR2 knock-out mice exhibited exacerbated immune pathologies in SARS-CoV[82]. These studies and our results suggest a protective function of CCR1 and CCR2 in early immune responses against coronaviruses. Both CCR1 and CCR2 interact with pro-inflammatory chemokines which are upregulated in the lungs of severe COVID-19 patients[63]. Thus, altered expression of CCR1 and CCR2 at steady state might influence the severity of COVID-19.

Type I interferon is crucial for antiviral immune responses and orchestrates the induction of chemokines and pro-inflammatory cytokines[28,31]. We observed reduced levels of pDCs, the main source of type I interferon during viral infections[36]. Thus, reduced frequencies of pDCs at baseline may contribute to the impaired or delayed type I interferon response in severe COVID-19[32,33]. Similar delayed type I interferon responses occur in SARS and MERS and are associated with worse disease outcome[83–85]. Therefore, dysregulated and delayed type I interferon response can be detrimental for the host in coronavirus infections.

On the contrary, we observed increased expression of IFNAR2 on basophils and myeloid cells but not on B cells and pDCs in individuals recovered from severe COVID-19 (Fig. 5d). This is in contradiction with inferences from a recent study which combined GWAS and bulk transcriptomics and identified reduced expression of IFNAR2 in lung and whole blood as a risk factor for severe COVID-19[2]. In contrast to bulk transcriptomics, we show at the single-cell level that IFNAR2 is

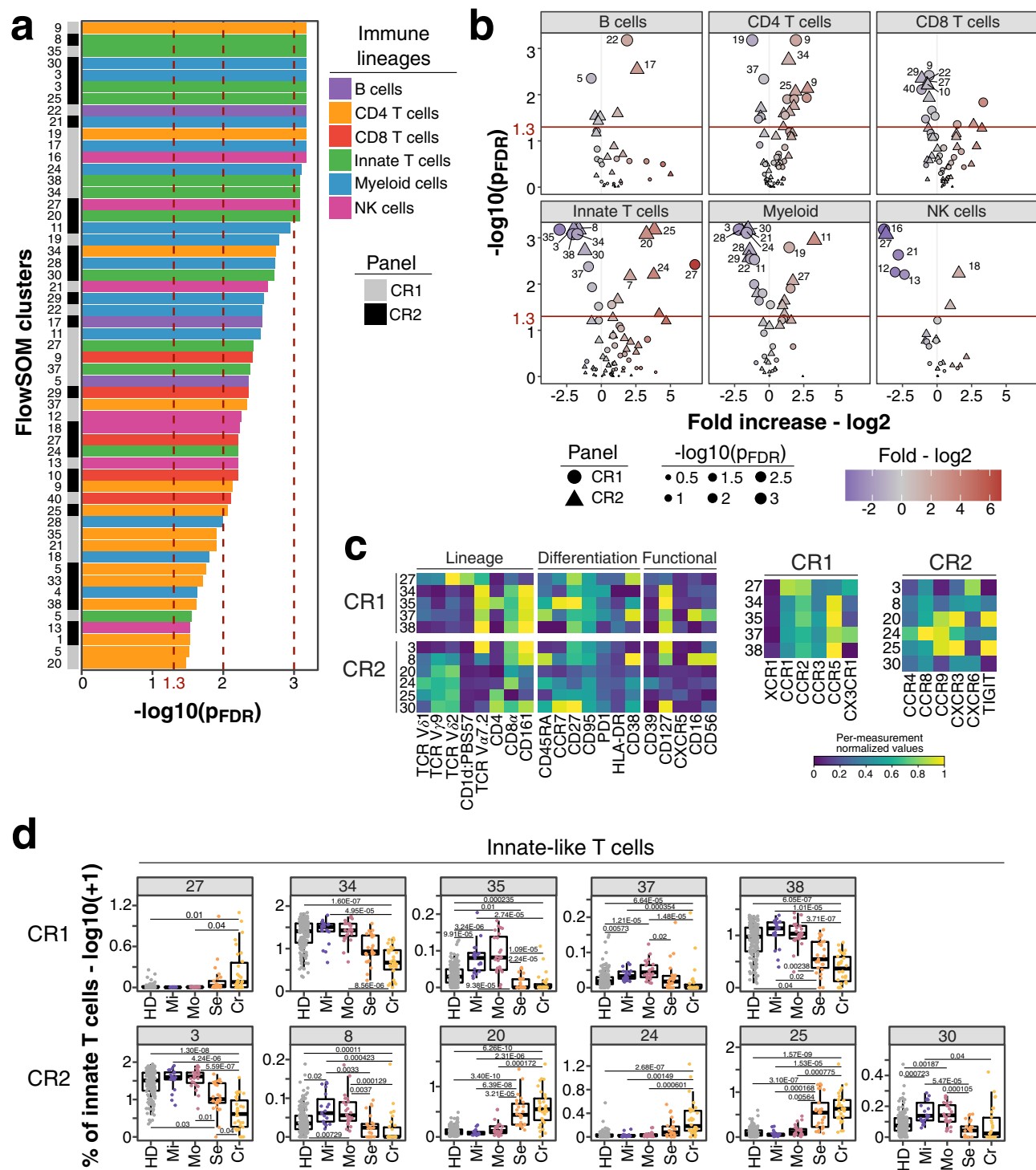

only affected on certain blood immune cell populations in individuals recovered from severe COVID-19. Notably, we measured IFNAR2 only on B cells, basophils and myeloid cells and can therefore not determine whether its expression is downregulated in other blood cell types. The dichotomy between reduced pDC frequencies and elevated IFNAR2 expression on myeloid cells is puzzling. However, interaction between type I interferon and its receptor results in endocytosis[86] and it is therefore possible that constitutively expressed type I interferon might regulate IFNAR2 expression at steady-state[87]. Nevertheless, the increased levels of IFNAR2 might potentiate the responsiveness of myeloid cells to type I interferon and thus drive exacerbated inflammation.

Most immune perturbations caused by COVID-19 disappear within 60 days post-infection, but some immune perturbations persist for weeks after viral clearance[37,38]. In our study, the majority of immune traits were at baseline in recovered patients (Fig. 1b). Nonetheless, we identified several immune traits which did not fully return to baseline even weeks after symptom onset (Figs. 1b, 2). Most of these long-term perturbations occurred in severe COVID-19, likely due to increased immune activation[57], and affected mainly B and T cells (Fig. 2). The half-life of peripheral lymphocytes is longer compared to myeloid cells. Models suggest that peripheral dendritic cells and monocytes are replenished every few days[40–42] while turnover of memory and naïve T cells can be in the order of several weeks[43] and

**Fig. 5 | Unsupervised analysis of immune system in individuals recovered from non-severe and severe COVID-19. a** FDR-adjusted −log10 *P*-values of FlowSOM clusters (*N* = 55) which differ significantly (*P* < 0.05) between individuals recovered from non-severe (mild/moderate) and severe (severe/critical) COVID-19 are shown. Bars are colored based on lineage (B cells, purple; CD4 T cells, orange; CD8 T cells, red; innate-like T cells, green; myeloid cells, blue; NK cells, pink). Bar on the left indicates whether traits originate from chemokine receptor panel 1 (CR1, gray) or 2 (CR2, black). **b** Volcano plots show FDR-adjusted −log10 *P*-values and log2 fold change derived from comparison of FlowSOM clusters between individuals recovered from non-severe and severe COVID-19 cases. Main lineages are depicted in separated plots and contain FlowSOM clusters from both panels CR1 (circle) and CR2 (triangle). Data point size corresponds to −log10 *P*-values and color indicates log2 fold change. *P*-values from Fig. 5a and b derive from logistic regression analysis with correction for age and experiment batch. *P*-values were corrected for multiple testing using Benjamini-Hochberg false discovery rate. **c** Heatmap depicts normalized median fluorescence intensity (MFI) values for lineage, differentiation and functional markers from top significant innate-like T cell clusters (Fig. 5a, *P* < 0.01). Values derived from CR1 (top) and CR2 (bottom) panels are separated. Heatmaps

on the right highlight expression of markers specific for CR1 and CR2 panels including chemokine receptors, co-stimulatory markers and IFNAR2. Values are normalized based on trimmed 1–99% percentile values. Complete heatmaps for all innate-like T cell clusters are shown in Supplementary Fig. 12a, b. **d** Frequencies for same clusters described in Fig. 5c are shown as boxplots based on study group. Values are log10(+1) transformed and plotted on linear scale. Logistic regression with correction for age and experiment batch was used to identify significant clusters between non-severe and severe COVID-19. Only FlowSOM clusters (*N* = 291) which did not show temporal changes within moderate and severe COVID-19 cases are shown as described in the Online methods section and results (Supplementary Fig. 13a). Residuals from linear regression between immune trait and age were used to calculate statistics on age-corrected data. ANOVA with subsequent Wilcoxon test and Bonferroni correction on residuals was performed for statistics highlighted in boxplots. Boxplots depict median and interquartile range (IQR) and length of whiskers is 1.5 times IQR. Study groups encompass healthy unexposed controls (HD) and individuals recovered from mild (Mi), moderate (Mo), severe (Se) and critical (Cr) COVID-19. Source data are provided as a Source Data file.

---

years[44], respectively. Thus, the prolonged immune cell half-life might interfere with the replacement of impaired lymphocytes after COVID-19 infection. Furthermore, naïve T cells are maintained by homeostatic proliferation while thymic output declines in aging[44], which might contribute to sustained immune perturbations.

Post-acute Sequelae of COVID-19 (PASC) occurs in a fraction of individuals resulting in persisting symptoms, but its occurrence did not correlate with disease severity[88]. Several factors early during acute infection are associated with the risk of developing PASC such as Epstein-Barr virus (EBV) viremia, auto-antibodies, and bystander activation of Cytomegalovirus (CMV)-specific T cells[88]. In contrast, the immune composition in the blood remained largely unaffected[88]. Noteworthy, immune signatures associated with PASC occurred in lungs but not in blood[89]. In our study, we can not delineate immune trajectories associated with PASC since such information was not collected in our cohorts. However, based on these studies the impact of PASC on the immune composition seems to be minor in recovered individuals.

In summary, we identified several single cell-based immune signatures associated with the development of severe COVID-19 outcome. We specifically identified components of innate immunity, NK cells and innate-like T cells which are important for the earliest events in orchestrating efficient immune responses and in the clearance of other pathogens potentially worsening the disease outcome. Our data support current clinical efforts to modulate immune cell trafficking using chemokine receptor inhibitors or administration of interferon to treat severe COVID-19 patients[63,90–92].

## Methods
### Samples
PBMC samples from 173 unexposed healthy individuals enrolled as part of the VRC clinical trial program served as control group. Convalescent samples from individuals recovered from mild and moderate COVID-19 were collected at the NIH (Mild, *N* = 14; Moderate, *N* = 10) and Evergreen in Washington State (Mild, *N* = 5; Moderate, *N* = 14). In addition, PBMCs from individuals recovered from severe (*N* = 25) and critical (*N* = 30) COVID-19 were obtained from Washington University. Distinction between severe and critical cases was based on required ventilation. All individuals from the mild and moderate group resolved symptoms by the time of sample collection while all individuals from the severe and critical groups showed at least substantial improvement of symptoms. Information about time between symptom onset and sample collection was unavailable for two samples from the mild COVID-19 group. Detailed demographics are shown in Supplementary Fig. 1 and Supplementary Table 1. Informed consent was obtained from individuals in compliance with IRB procedures from the National

Institutes of Health (Bethesda, USA) and Washington University (Missouri, USA). Peripheral blood mononuclear cells (PBMC) were purified using density gradient centrifugation and cryopreserved in 10% DMSO in liquid nitrogen. In this cross-sectional study, one sample per individual was measured. All samples were collected prior to the availability of vaccines against SARS-CoV2. Furthermore, samples from the healthy control group were collected prior to the emergence of SARS-CoV2 and thus represent unexposed healthy individuals.

### Flow cytometry
Detailed information of buffers and cell culture media is listed in Supplementary Table 6 and staining reagents are listed in Supplementary Table 3. Staining reagents included Viability UV Blue (Manufacturer: Thermo Fisher Scientific, Cat#: L34962, titer per 50 µl staining volume: 0.0641 µl), anti-TCR Vd1 FITC (clone: TS8.2, Manufacturer: Thermo Fisher Scientific, Cat#: TCR2730, titer per 50µl staining volume: 2.5 µl), anti-CD127 BB630 (clone: HIL-7R-M21, Manufacturer: BD Biosciences, Cat#: 624294, titer per 50 µl staining volume: 1.25 µl), anti-PD-1 BB660 (clone: EH12.1, Manufacturer: BD Biosciences, Cat#: 624295, titer per 50 µl staining volume: 0.31 µl), anti-CD16 BB700 (clone: 3G8, Manufacturer: BD Biosciences (OptiBuild), Cat#: 746199, titer per 50 µl staining volume: 0.04 µl), anti-CXCR5 BB790 (clone: RF8B2, Manufacturer: BD Biosciences, Cat#: 624296, titer per 50 µl staining volume: 0.04 µl), anti-TCR Vg9 PE (clone: B3, Manufacturer: BD Biosciences, Cat#: 555733, titer per 50 µl staining volume: 1.25 µl), anti-TCR Vd2 PE-CF594 (clone: B6, Manufacturer: BD Biosciences, Cat#: 624352, titer per 50 µl staining volume: 0.01 µl), anti-CD161 PE-Cy5 (clone: DX12, Manufacturer: BD Biosciences, Cat#: 551138, titer per 50 µl staining volume: 2.5 µl), anti-HLA-DR PE-Cy5.5 (clone: TU36, Manufacturer: Thermo Fisher Scientific, Cat#: MHLDR18, titer per 50 µl staining volume: 0.31 µl), anti-CD1d:PBS57 tetramer APC (clone: -, Manufacturer: NIH tetramer core, Cat#: 41386, titer per 50 µl staining volume: 0.15 µl), anti-CD45RA Ax700 (clone: HI100, Manufacturer: BD Biosciences, Cat#: 560673, titer per 50ul staining volume: 0.63 µl), anti-CCR7 BUV496 (clone: 2-L1-A, Manufacturer: BD Biosciences, Cat#: 749827, titer per 50 µl staining volume: 5 µl), anti-CD56 BUV563 (clone: NCAM16.2, Manufacturer: BD Biosciences, Cat#: 565704, titer per 50 µl staining volume: 0.31 µl), anti-CD39 BUV661 (clone: TU66, Manufacturer: BD Biosciences, Cat#: 749967, titer per 50 µl staining volume: 0.63 µl), anti-CD95 BUV737 (clone: DX27, Manufacturer: BD Biosciences, Cat#: 624286, titer per 50 µl staining volume: 1.25 µl), anti-CD4 BUV805 (clone: SK3, Manufacturer: BD Biosciences, Cat#: 564910, titer per 50ul staining volume: 0.63µl), anti-CD3 BV510 (clone: UCHT1, Manufacturer: BD Biosciences, Cat#: 563109, titer per 50µl staining volume: 0.15µl), anti-CD8a BV570 (clone: RPA-T8, Manufacturer: Biolegend, Cat#: 301038, titer per 50µl staining volume: 0.267µl), anti-

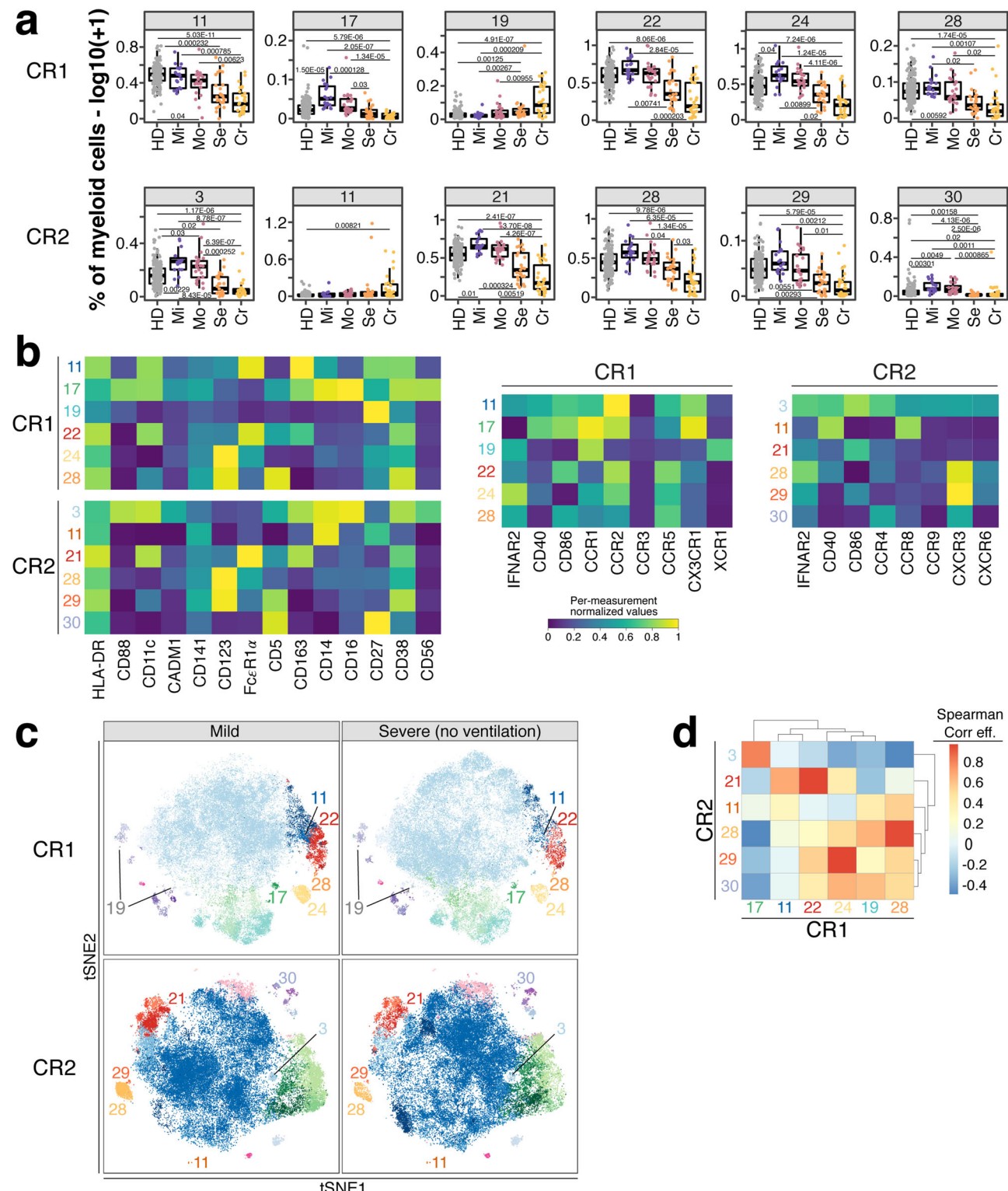

CD38 BV605 (clone: HIT2, Manufacturer: BD Biosciences, Cat#: 740401, titer per 50ul staining volume: 1.25µl), anti-TCR Va7.2 BV711 (clone: 3C10, Manufacturer: Biolegend, Cat#: 351732, titer per 50µl staining volume: 1.25µl), anti-CD27 BV786 (clone: L128, Manufacturer: BD Biosciences, Cat#: 624292, titer per 50µl staining volume: 0.31µl), anti-CADM1 FITC (clone: 30, Manufacturer: MBL International Corporation, Cat#: CM004-4, titer per 50µl staining volume: 0.15µl), anti-CD141 BB630 (clone: 1A4, Manufacturer: BD Biosciences, Cat#: 624294, titer per 50µl staining volume: 0.01µl), anti-CD123 BB660 (clone: 7G3, Manufacturer: BD Biosciences, Cat#: 624295, titer per 50ul staining volume: 0.04µl), anti-FcEr1a BB700 (clone: AER-37, Manufacturer: BD Biosciences, Cat#: 747780, titer per 50µl staining volume: 0.63µl), anti-IgD BB790 (clone: IA6-2, Manufacturer: BD Biosciences, Cat#: 624296, titer per 50µl staining volume: 0.31µl), anti-IFNAR2 PE (clone: REA124, Manufacturer: Miltenyi, Cat#: 130-099-555, titer per 50µl staining volume: 1.25µl), anti-CD88 PE-Dazzle594 (clone: S5/1, Manufacturer: Biolegend, Cat#: 344318, titer per 50µl staining volume: 0.31µl), anti-CD3 PE-Cy5 (clone: UCHT1, Manufacturer: BD Biosciences, Cat#: 555334, titer per 50µl staining volume: 0.31µl), anti-CD5 PE-Cy5.5 (clone: CD5-5D7, Manufacturer: Thermo Fisher, Cat#: MHCD0518, titer

**Fig. 6 | Myeloid cell populations from FlowSOM analysis as potential predictor for disease outcome. a** Frequencies (log10 + 1) of myeloid cell clusters among top hits ($P < 0.01$) described in Fig. 5a are shown for CR1 (top row) and CR2 panel (bottom row). Values are log10(+1) transformed and plotted on linear scale. Residuals from linear regression between immune trait and age were used to calculate statistics on age-corrected data. ANOVA with subsequent two-sided Wilcoxon test and Bonferroni correction on residuals was performed for statistics highlighted in boxplots. Boxplots depict median and interquartile range (IQR) and length of whiskers is 1.5 times IQR. Study groups encompass healthy unexposed controls (HD) and individuals recovered from mild (Mi), moderate (Mo), severe (Se) and critical (Cr) COVID-19. **b** Heatmaps showing normalized median fluorescence intensity (MFI) values for clusters described in Fig. 6a are shown. Heatmaps on the left show markers used to delineate immune cell subsets. On the right, heatmaps depict CR panel-specific markers. Values are normalized based on trimmed 1–99% percentile values. **c** tSNE plots with myeloid cells from individuals recovered from mild (left column) or severe (right column) COVID-19 are shown. Data from panels CR1 and CR2 are shown in the top and bottom row, respectively. Each plot contains 50,000 subsampled myeloid cells (gating shown in Supplementary Fig. 2d). Dots are colored based on FlowSOM cluster annotation and full data is shown in Supplementary Fig. 10a–c. Clusters described in Fig. 6a and b are annotated and highlighted. **d** Spearman analysis of normalized MFI values between clusters described in Fig. 6a is shown in order to estimate the phenotypic overlap between CR1 and CR2 panel. Heatmap depicts Spearman correlation coefficient.

per 50 µl staining volume: 0.31 µl), anti-CD11c APC (clone: B-ly6, Manufacturer: BD Biosciences, Cat#: 559877, titer per 50 µl staining volume: 5 µl), anti-CD27 APC-R700 (clone: M-T271, Manufacturer: BD Biosciences, Cat#: 624348, titer per 50 µl staining volume: 0.63 µl), anti-CD40 BUV496 (clone: 5C3, Manufacturer: BD Biosciences, Cat#: 741159, titer per 50 µl staining volume: 1.25 µl), anti-CD56 BUV563 (clone: NCAM16.2, Manufacturer: BD Biosciences, Cat#: 565704, titer per 50 µl staining volume: 0.31 µl), anti-CD21 BUV661 (clone: B-ly4, Manufacturer: BD Biosciences, Cat#: 741605, titer per 50 µl staining volume: 0.31 µl), anti-CD163 BUV737 (clone: GHI/61, Manufacturer: BD Biosciences, Cat#: 741863, titer per 50 µl staining volume: 5 µl), anti-CD20 BUV805 (clone: 2H7, Manufacturer: BD Biosciences, Cat#: 612905, titer per 50 µl staining volume: 2.5 µl), anti-CD14 BV510 (clone: MPhiP9, Manufacturer: BD Biosciences, Cat#: 624289, titer per 50 µl staining volume: 0.1 µl), anti-CD16 BV570 (clone: 3G8, Manufacturer: Biolegend, Cat#: 302036, titer per 50 µl staining volume: 1.25 µl), anti-CD38 BV605 (clone: HIT2, Manufacturer: BD Biosciences, Cat#: 740401, titer per 50 µl staining volume: 1.25 µl), anti-CD86 BV711 (clone: 2331, Manufacturer: BD Biosciences, Cat#: 563158, titer per 50 µl staining volume: 0.63 µl), anti-HLA-DR BV786 (clone: G46-6, Manufacturer: BD Biosciences, Cat#: 564041, titer per 50 µl staining volume: 0.15 µl), anti-CCR2 BV421 (clone: 48607, Manufacturer: BD Biosciences, Cat#: 564067, titer per 50 µl staining volume: 5 µl), anti-CCR3 BUV395 (clone: 5E8, Manufacturer: BD Biosciences, Cat#: 743063, titer per 50 µl staining volume: 2.5 µl), anti-CCR5 BV650 (clone: 2D7/CCR5, Manufacturer: BD Biosciences, Cat#: 740600, titer per 50 µl staining volume: 2.5 µl), anti-CX3CR1 BV750 (clone: 2A9-1, Manufacturer: BD Biosciences, Cat#: 747376, titer per 50 µl staining volume: 5 µl), anti-CCR1 PE-Cy7 (clone: 5F10B29, Manufacturer: Biolegend, Cat#: 362914, titer per 50 µl staining volume: 5 µl), anti-XCR1 APC-Fire750 (clone: S15046E, Manufacturer: Biolegend, Cat#: 372608, titer per 50 ul staining volume: 5 µl), anti-CCR9 BV421 (clone: L053E8, Manufacturer: Biolegend, Cat#: 358914, titer per 50 µl staining volume: 5 µl), anti-CCR8 BUV395 (clone: 433H, Manufacturer: BD Biosciences, Cat#: 747573, titer per 50 µl staining volume: 1.25 µl), anti-CCR4 BV650 (clone: 1G1, Manufacturer: BD Biosciences, Cat#: 744140, titer per 50 µl staining volume: 5 µl), anti-CXCR6 BV750 (clone: 13B 1E5, Manufacturer: BD Biosciences, Cat#: 747052, titer per 50 ul staining volume: 5 µl), anti-CXCR3 PE-Cy7 (clone: G025H7, Manufacturer: Biolegend, Cat#: 353720, titer per 50 µl staining volume: 1.25 µl), anti-CD19 APC-H7 (clone: SC25C1, Manufacturer: BD Biosciences, Cat#: 560177, titer per 50 µl staining volume: 0.31 µl) and anti-TIGIT APC-Cy7 (clone: A15153G, Manufacturer: Biolegend, Cat#: 372734, titer per 50 µl staining volume: 0.63 µl).

Staining reagent cocktails were prepared in staining buffer (RPMI without phenol red and 4% HINCS) containing Brilliant Buffer Plus (1:5 diluted) and TrueStain Monocyte Blocker (5 µl/100 µl). Antibody cocktails were tested on irrelevant PBMC sample to validate completeness prior to sample processing. After successful validation of staining reagent cocktails, PBMCs were thawed in RPMI containing 10% fetal bovine serum, 100 IU/ml Penicillin, 100 µg/ml Streptomycin and 292 µg/ml L-Glutamine (referred to as R10) containing 50 U/ml Benzonase using a tube adaptor to facilitate and standardize the thawing process as described[93]. Cells were washed once with 5 ml R10 and transferred to a V-bottom, 96-well plate (Corning). After two washes with 200 µl PBS, cells were stained in 100 µl fixable Live/Dead Blue viability dye containing human BD Fc receptor block (5 µl/100 µl) for 20 minutes at room temperature protected from light. Afterwards, cells were distributed into two 96-V bottom plates and stained with 50 µl of either B cell/myeloid cell (BDC) or T cell/NK cell (TNK) backbone staining mix for 30 min at room temperature. Samples were subsequently distributed into two wells and stained with either chemokine receptor panel 1 (CR1) or 2 (CR2) for 30 min at room temperature. Subsequently, we washed cells three times with 250 µl staining buffer followed by fixation with 0.5% paraformaldehyde in PBS overnight at 4 C. Cells were acquired the next day with a FACSymphony (BD Biosciences) cytometer using FACS DIVA software (version 9.1). Detailed instrument configuration is described elsewhere[94]. Initial centrifugation for thawing was performed at $700 \times g$ for 5 min and all subsequent centrifugation steps were done at $860 \times g$ for 3 min.

Samples were processed in two batches and samples from the different cohorts/study groups were equally distributed across the two experiments to mitigate potential issues with batch effects. PBMCs from the same blood draw and batch from an unexposed healthy individual was measured in both experiments to assess reproducibility.

## Data analysis

Irregular events and outliers in the raw data were determined and excluded using R-implemented (R version 4.0.0) FlowAI (version 1.18.5)[95]. Subsequently, correction for spectral overlap (compensation) was performed in FlowJo 10.1.7 (BD Biosciences) using single-stained beads. A new set of fcs files only containing viable, high-quality (based on FlowAI) cells was generated for subsequent analysis of immune cell traits with FlowJo 10.1.7. For the BDC-CR1 panel, gates from two donors required adjustments due to slight signal shifts caused by irregularities in data acquisition which were not detected by FlowAI. Otherwise, identical gates were used across all samples and batches. Markers were divided in two groups based on their purpose to either define immune cell subsets or functional markers/characteristics (Supplementary Table 2). Three different parameters were extracted for subsequent analysis, namely frequency of immune cell populations, frequency of cells expressing functional markers and mean fluorescence signal. Biologically relevant expression was assessed and immune traits with insufficient frequencies or irrelevant expression patterns were manually excluded which resulted in 1758 out of 3787 manually defined immune traits.

For tSNE and FlowSOM analysis, CD4$^+$ T cells and CD8$^+$ T cells (both gated from CD3$^+$CD4$^+$Vγ9$^-$Vδ1$^-$Vδ2$^-$CD1d:PBS57$^-$ conventional T cells), B cells (HLA-DR$^+$CD20$^+$), myeloid cells (HLA-DR$^+$CD20$^-$), innate-like T cells (NKT cells, MAIT cells and cells positive for TCR-γ or -δ reagents) and NK cells/innate lymphoid cells (CD3$^-$HLA-DR$^-$) were separately concatenated from the two chemokine receptor panels CR1 and CR2. The same individuals were included as described for the manual gating analysis, with the exception that we excluded the two

samples from the BDC-CR1 panel data which had slight signal shifts as described above. Subsequently, dye aggregates were removed by manual gating to avoid artefacts. The cleaned events were exported as new fcs files and used for R-implemented tSNE (Rtsne, version 0.15) and FlowSOM (version 1.20.0). For FlowSOM, 40 clusters were defined for CD4+ and CD8+ T cells and 30 clusters for all other immune subsets. For clustering, markers used to initially define and extract these immune subsets were excluded from the clustering analysis (Supplementary Table 4). We excluded these markers to avoid parsing of background signal or uniform expression into artificial subpopulations[96]. Clusters with unusual expression pattern occurred likely because of residual immune cell contaminations and were removed from downstream analysis (Supplementary Table 5). Raw data output from FlowSOM and tSNE analysis are visualized in Supplementary Figs. 10–12. Subsequent analysis was performed with remaining 388 FlowSOM clusters (B cells, $N = 60$; myeloid cells, $N = 60$; innate-like T cells, $N = 78$; conventional CD4 T cells, $N = 76$; conventional CD8 T cells, $N = 76$; and NK cells, $N = 38$). For tSNE, 50,000 cells (27,583 cells for innate-like T cells and 25,000 for NK cells) from each severity group were concatenated prior to tSNE analysis (perplexity = 30, theta = 0.5, 5000 iterations) in order to maintain priority for tSNE computation equal among patient groups. Fewer cells were used for TSNE in the case of innate-like T cells due to limited numbers of cells in the critical COVID-19 group (total 27,583 cells from all patients). We expected lower diversity of NK cell subsets and therefore used 25,000 cells per study group for tSNE.

### Statistical analysis

**Exclusion of individuals.** Samples with considerable number of missing manually defined immune trait values were excluded using the missCompare (version 1.0.3) package in R. A cut-off of 10% was applied (i.e., samples with more than 10% missing values were excluded). Two individuals were excluded based on missingness of values for immune traits. None of the immune traits were excluded based on missingness (Cut-off of 80% missing values). For FlowSOM analysis, same samples were used according to the missingness analysis on manually defined traits. Of note, the FlowSOM model was trained on all samples irrespective of missingness to ensure maximum number of cells per study group to train the FlowSOM model.

**Assessment of long-term immune perturbations.** We distinguished immune traits which were affected by long-term immune perturbations or at steady-state within moderate and severe COVID-19 group. We focused on these two study groups because they span across the longest period between symptom onset and sample collection enabling the most precise analysis of long-term immune trajectories after symptom onset (Supplementary Fig. 1a). Of note, age correlated with hospitalization length in severe but not critical cases and was significantly shorter in severe COVID-19 cases (Supplementary Fig. 1b, c). We used linear regression between rank-normalized immune traits derived from both unsupervised clustering and manual analysis and length of time in days between symptom onset and sample collection. In addition, we compared immune traits in samples with less or more than 60 days between symptom onset and sample collection using Wilcoxon signed-rank test. Long-term perturbated traits were defined as manually defined immune traits with unadjusted $P < 0.001$ in at least one of the analyses ($N = 24$).

**Identification of immune traits predictive for COVID-19 severity.** Immune traits and FlowSOM clusters with unadjusted $P > 0.05$ in both analyses described above (linear regression and Wilcoxon signed-rank test) were defined as stable immune traits at steady-state (1365 manually defined immune traits and 291 FlowSOM clusters) and were used to predict immune signatures associated with the development of severe COVID-19. We rank-normalized the data and used logistic

regression between mild/moderate (group non-severe) and severe/critical (group severe) cases and corrected for age and experiment (batch). $P$-values were adjusted using Benjamini-Hochberg false discovery rate[97] and adjusted $P$-values <0.05 were considered statistically significant.

We ran the logistic regression analysis with all individuals ($N = 98$) or only individuals who were not hospitalized or discharged at day of sample collection ($N = 71$) since hospitalizations at sample collection might point towards persisting symptoms and immune perturbations and affect our analysis (Supplementary Fig. 9c, d). For overall assessment of significant difference of immune traits and FlowSOM clusters between study groups we used one-way ANOVA on age-corrected residuals derived from linear regression on rank-normalized data. Subsequently, if one-way ANOVA results were significant, we used two-sided Wilcoxon test with Bonferroni correction for comparison of study groups.

### Reporting summary

Further information on research design is available in the Nature Portfolio Reporting Summary linked to this article.

## Data availability

The Flow Cytometry data generated in this study have been deposited in the FlowRepository database under accession code FCM-Z5PC. The raw data files contain pre-gated viable cells. Source data are provided with this paper.

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

## Acknowledgements

This work was supported by the intramural research program of the Vaccine Research Center (NIAID. NIH) to M.R. This study utilized samples obtained from the Washington University School of Medicine's COVID-19 biorepository, which is supported by: the Barnes-Jewish Hospital Foundation; the Siteman Cancer Center grant P30 CA091842 from the National Cancer Institute of the National Institutes of Health; and the Washington University Institute of Clinical and Translational Sciences grant UL1TR002345 from the National Center for Advancing Translational Sciences of the National Institutes of Health. The content is solely the responsibility of the authors and does not necessarily represent the view of the NIH. M.M. is supported by the National Institute for Health Research (NIHR)-funded BioResource, Clinical Research Facility and Biomedical Research Centre based at Guy's and St Thomas' NHS Foundation Trust in partnership with King's College London. T.L. is supported by the International Society for Advancement of Cytometry (ISAC) Marylou Ingram Scholar Program. We thank the Flow Cytometry Facility at the Vaccine Research Center including Erica Smit, Esther Thang, Richard Nguyen and Steve Perfetto. In addition, we thank Ingelise Gordon, Charla Andrews, Maria Burgos Florez, Laura Novik, Britta Flach, Emily Coates, Nina Berkowitz and Martin Gaudinski from the VRC Clinical Trials Program and Obrimpong Amoa-Awua and Adrian McDermott from the Vaccine Immunology Program for their support regarding PBMC samples from the NIH and Evergreen cohort. Furthermore, we thank all donors from the VRC and the Washington University cohorts.

## Author contributions

T.L. and M.R. designed the study; T.L., Y.I., and M.B. performed all experiments; P.M., C.W.G., and J.A.O. provided patient samples; T.L. and M.R. performed data analysis; M.M. supported statistical analysis; T.L. and M.R. wrote the manuscript. All authors reviewed and approved the manuscript.

## Funding

## Competing interests

The authors declare no competing interests.
