## [Peer Review File · Nature Communications]

Immune phenotypes that are associated with subsequent
COVID-19 severity inferred from post-recovery
samplesREVIEWER COMMENTS

Reviewer #1 (Remarks to the Author):

Liechti and colleagues report on immune phenotypes that predict COVID-19 severity using PBMC samples from 98 patients post-recovery (defined as/assumed steady-state levels) and 173 healthy controls. Mild and moderate patients resolved symptoms by the time of sample collection (medians of 40 and 45 days from onset of symptoms respectively), while patients from the severe and critical groups (77 and 33 days median from onset of symptoms, respectively) showed at least substantial improvement of symptoms.

The authors measure the immune composition of peripheral blood using high-dimensional flow cytometry and they identify distinct chemokine receptor signatures: individuals recovered from severe COVID-19 had increased expression of lung-homing chemokine receptors CCR4, CXCR3, and CX3CR1 on NK cell subsets, and homing-receptor CCR4 on trans. mem. CD4+ and CD8+ T cells, which can result in exacerbated lung inflammation. In contrast, higher levels of TIGIT were found in mild and moderate group patients, potentially actively preventing immune pathologies and severe lung damage. In addition, CCR1 and CCR2 were reduced on monocyte subsets in severe cases, potentially associated with a protective role.

The findings of the authors will be of significance to the field and the approach of trying to find predictive markers using post-symptomatic samples is quite unique and inventive (though challenging).

The manuscript is well written, well-structured, and the statistical methods seem sound.

I would like to address the following comments:

Major points:

1)

a) Can the authors elaborate more on why it would be reasonable to assume that these post-recovery samples are proper representatives for pre-COVID-19 samples? Most critical patients were still hospitalized and sampled very much earlier than the other patients.

The authors seem to have dealt with this to a certain extent by leaving out the critical group in their analysis at page 6 line 130 (inclusion of only moderate and severe group). At page 8 line 190 they take another approach to correct for this effect: they include only the discharged individuals which means that they have included 8 critical patients and excluded 6 severe patients (extended data fig 1). Why do the authors change their approach to correct for this potential bias and what would happen if they compare only moderate and severe groups for this analysis at page 8?

Fig 2d exemplifies the potential bias that may have resulted from the difference in day of sampling between the groups (severe versus moderate in this case): severe patients at day 100 post onset have higher frequency % than moderate patients at day 100 whereas severe patients at day 40 had lower levels than moderate patients at day 40. Thus, the composition of the groups (sampling date post onset of symptoms) was essential and may have influenced the other analyses (page 6, line 130).

The reviewer understands that this was a factor the authors had no influence on but can the authors comment on this in the manuscript where they have taken this into account in their analyses (for example by comparing only samples taken in a similar time range). At least, the authors can comment in the discussion on how this may have influenced their results.

b) In addition, the authors mention they aim to “identify immune perturbations which may contribute to long COVID-19” (line 128 page 6), however it is not very clear if these type of symptomatology is taken into account in their analyses. Potentially, this may even have affected their results, when patients with and without long-COVID-19 symptomatology are considered as one group when studying their immunological profile. Could the authors elaborate on this in the text as well?

c) in the abstract the authors write: line 29 “focused only on those immune signatures returning to steady-state.” This may need some clarification, since different approaches are followed in different

analyses in different paragraphs of the result and it does not seem the case that the authors have included only parameters that returned to steady-state levels in all groups. Please clarify.

2)

The findings on the naïve CD4+ T cells and naïve CD8 cells (fig 2c and 3c) are significant and of interest to the field. Can the authors elaborate on those findings in the discussion and describe how this relates to the findings of others published in literature to date?

Minor points:

3)

With regards to the controls, could the authors describe the vaccination status or year of inclusion? Can vaccination prior to sampling have affected the data?

4)

In the top paragraph at page 7 the authors describe “the expression levels over time”. Can the authors for clarity write there and/or in the legend that each patient is sampled only once? Just to distinguish from temporal dynamics analysis within patients.

Reviewer #2 (Remarks to the Author):

The authors describe differences in steady state immune phenotypes after mild and severe COVID-19. The methods and results are well described, but the conclusion drawn is not sound.

The large number in each group provides confidence in the results and will allow future studies to understand both recovery and potentially also an association with long COVID. The findings of note include reduced levels of chemokine receptors on monocyte subsets and reduced frequencies of type I IFN-producing pDCs, demonstrating differences in cell types that play a role in the earliest host response to infection. However, the authors have overinterpreted the findings as phenotypes that PREDICT COVID19 severity – this cannot be the case given the findings (some referenced by the authors) that immunologic perturbations persist after recovery. Many of the differences identified may very well be a result of mechanisms to reset/regulate the host’s innate response and reduce the inflammatory response if the host were re-infected. It is essential that the pre-infection immune phenotype of mild and severe COVID19 cases be compared to draw conclusions regarding phenotypes that predict COVID19 severity. The authors state on line 177 that “we hypothesized that stable immune traits between symptom onset and sample collected remained at or returned early to pre-infection baseline” but instead of providing data to support this hypothesis, it appears that the authors have assumed it to be true.

The title, abstract and text that refer to the results “predicting” COVID19 severity should be modified to reflect that the results are associated with severe COVID19. Alternatively, data to demonstrate the phenotypes associated with severe COVID19 are indeed present prior to infection, could be added. Perhaps there is information regarding COVID19 infection for some of the comparators (healthy individuals) that would provide a source of these data.

As written, the manuscript describes traits at “baseline” (line 400/401) for data used to generate Figure 1b from analyzing differences explained as “before and after 60 days of symptom onset” on line 909; the methods describe these as “convalescent” and state that the PBMCs from those in severe and critical cases were from individuals who were recovered (line 432). Plots shown in Figure 2 imply that the earliest time points were 2-3 weeks after symptom onset. In looking at this information again, I realize that I’ve misunderstood that samples were collected from the same individual at different times post infection. The methods as well as text should plainly state that single samples from individuals who had COVID19 symptoms of various degrees were used in the analysis of changes over time, or provide the specific consecutive days post-illness onset together with the

change over time.

Some of the data presented may mislead readers who are looking at images only, as it appears different cell numbers are presented in the figures; an example is Figure 3d; while the % of cells expressing TIGIT is clearly stated, it would help to either present the same number of cells in the plots or to state the number of cells plotted, in the figure legend.

Reviewer #3 (Remarks to the Author):

This manuscript by Leichti et al. aimed to determine the unique immune signatures that predispose to severe COVID-19. They assessed peripheral blood immune cells and cytokine receptor expression profiles in subjects recovered from mild, moderate, severe, and critical COVID-19 diseases. They proposed that measuring immune composition in recovered subjects when immune responses restore back to baseline would provide insight in the key and differential factors that drive severe diseases. By using 28-color high dimensional flow cytometry, they found that those who recovered from severe COVID-19 had elevated expressions of lung-homing receptors CCR4, CXCR3 and CV3CR1 on NK cells, and also had increased CCR4 expression on memory CD4 and CD8 T cells. CD8 T cells (central memory and effector T cells) also showed reduced levels of TIGIT, which had been shown to dampen disease pathogenesis caused by influenza. The number of central memory CD4+ and CD8+ T cells were reduced and CD8+ effector and terminal memory T cells were increased in those recovered from severe disease. MAIT cells were also reduced in frequency but with increased TIGIT expression.

Monocytes also displayed decreased CCR1 and CCR2 expressions that have been reported to dampen SARS-CoV-2-induced inflammation in vivo. These results indicate excessive homing of NK and T cells into the lung, possibly driving excessive inflammation and tissue damage. Plasmacytoid dendritic cells (pDCs) are the primary source of antiviral cytokines type I interferons (IFNs), but the authors show pDC numbers were lower in those who recovered from severe COVID-19. IFN receptor (IFNAR2) expression was increased on pDCs and monocytes from those recovered from severe COVID-19, possibly indicating reduced antiviral responses due to reduced productions of type I IFNs by pDCs despite increased IFNAR2 expression. The authors concluded that these immune features after recovery from severe COVID-19 reflect the immune system at steady-state and are comparable to pre-infection and these immune signatures predict the trajectory of disease severity.

This study provides novel immunological insights in COVID-19, especially the NK homing, TIGIT expressions on T/MAIT cells as well as IFNAR2 expressions, and while the flow cytometry is mostly methodologically sound, there are key questions on donor characteristics, vaccine status and other potential confounding factors that need to be included to solidify authors claims

General comments

Although a reference has been provided for the statement that human immune system rapidly reverts to steady-state with a composition comparable prior to infection (line 336-338), it is more appropriate to describe this as recovered state instead of steady-state. Despite the authors claim and the reference given, the presence of possible post-COVID syndrome or long-COVID that persists months after acquiring COVID may be an important confounder and needs to be carefully considered and examined in this study.

Would the existing medical conditions such as obesity, asthma, or COPD impact the identified immune traits identified in this study ? were these info recorded ?

Were vaccination status recorded for all recruited participants and how would this impact this results and conclusions of this study.

The results section particularly when describing cell phenotypes and their surface receptors, it is important to also describe what these receptors do in the context of cell recruitment/activation and in infection/immunity for better clarity to the readers.

Please discuss in the discussion section on what do the reduced or increased assessed immune cells and their activation or receptor expression (that were assessed in this study) mean for the immune responses and disease severity. le What does reduced MAIT and other cells mean in terms of immune responses? More inflammation and less tissue repair ?

Specific comments

Line 99, PBMC from 173 healthy individuals... it is more appropriate to indicate that they are recovered individuals.

Line 100, please include donor demographics and characteristics.

Line 104, please indicate that these 4 unique sets of markers are.

Extended data fig 1a and line 132, what does the y-axis represent? and the numbers below the box of the boxplot ? median day of symptom onset and sample collection ? why is this shorter for critical conditions requiring ventilation vs other groups?

Throughout the manuscript it is stated that PBMCs were obtained from recovered individuals, but for Fig 2 b-f there are data presented as “healthy”, were these data compared to recovered subjects and different disease severity?

Line 153 – 156, need to state what CD38+ and CD38- represents ? T cell activation ? it should also include that it was the “early” decrease in CD38+ and decrease in CD38- naïve T cells that were associated with severe diseases.

Line 163, given basophil CCR3 expression and IFNAR2 (line 382) was altered in those recovered from severe COVID-19, was basophil frequency also altered ?

Line 188, patients with severe and critical disease were still hospitalized at the time of sample collection, and their data was excluded and those who were discharged were included in the data analyses. Would it be interesting to look at whether the assessed parameters are also altered in this group vs other groups th

Line 196, non-hospitalized patients refer to discharged or those requiring no hospitalization ?

Line 194, “65 significant hits were shared between...” and “only 6 new immune traits were discovered “

Please describe what those hits are and in addition to the venn diagram are these hits presented in the figs at all ?

Line 249, “the chemokine receptor profile on dendritic cells did not differ substantially between...” please indicate the fig this is referred to.

Line 263 – 264, “IFNAR2 expression was lowest on naïve B cells and highest on pDCs and cDC1s...”, and in the next sentence “ we found increased expression of IFNAR2 on monocyte and dendritic cell subsets, except for cDC1s and pDCs.” These two sentences conflict with each other.

Was IFNAR2 expression tracked as in Fig2e on these cells?

Fig 4d, the line graph beside the flow histogram, as the data is not continuous, perhaps this is best presented as bar graph.

Line 280, would be interesting to look at whether the differentially expressed immune signatures assessed are also altered in the immune perturbed group that was excluded from the study and whether these altered immune phenotypes also contribute to the disease severity.

Line 289, "from each chemokine receptor panels... 5 and 6 significant clusters resemble innate-like T cells" please indicate where they are in fig 5b.

Line 297 and 298, should the fig referred in the text (Fig. 5d) be Fig 5c?

Please indicate what type of T cells V δ 2V γ 9 refers to.

Are NK cell numbers increased or decreased in different disease states?

Reductions in cell frequency in peripheral blood such as cDC1s early in days after symptom onsets, could this be due to infiltration of cDC1s into the lung (thus a reduction in blood)? And therefore indicate early antigen presentation? why weren't all cell types and surface receptors tracked through days after symptom onsets (like in Fig2 for B, cDCs, and basophils)? ie CD8 effector T and MAIT cells, monocytes?

REVIEWER COMMENTS

We thank the reviewers for their careful and critical reading of our manuscript. This review has certainly strengthened our presentation! We addressed the comments of all reviewers, summarized below. We begin with the common concern raised by all three reviewers, and then address the individual criticisms point-by-point. In black font is the reviewer text, copied verbatim, followed by our response in red font and highlighting our changes to the text in blue font.

Comments/concerns raised by all three reviewers

Reviewer #1

Can the authors elaborate more on why it would be reasonable to assume that these post-recovery samples are proper representatives for pre-COVID-19 samples? Most critical patients were still hospitalized and sampled very much earlier than the other patients.

Reviewer #2

However, the authors have overinterpreted the findings as phenotypes that PREDICT COVID19 severity – this cannot be the case given the findings (some referenced by the authors) that immunologic perturbations persist after recovery. Many of the differences identified may very well be a result of mechanisms to reset/regulate the host's innate response and reduce the inflammatory response if the host were re-infected. It is essential that the pre-infection immune phenotype of mild and severe COVID19 cases be compared to draw conclusions regarding phenotypes that predict COVID19 severity. The authors state on line 177 that “we hypothesized that stable immune traits between symptom onset and sample collected remained at or returned early to pre-infection baseline” but instead of providing data to support this hypothesis, it appears that the authors have assumed it to be true.

Reviewer #3

Although a reference has been provided for the statement that human immune system rapidly reverts to steady-state with a composition comparable prior to infection (line 336-338), it is more appropriate to describe this as recovered state instead of steady-state. Despite the authors claim and the reference given, the presence of possible post-COVID syndrome or long-COVID that persists months after acquiring COVID may be an important confounder and needs to be carefully considered and examined in this study.

Reply

We thank all three reviewers for their comments regarding our analysis. Our analysis relies on published observations, that in general, the immune system reverts to baseline upon pathogen clearance, commonly referred to as immune contraction (1). This has been demonstrated for numerous acute viral infections such as Ebola (2), Zika (3), Dengue (4, 5), measles (6), hantavirus (7) and respiratory syncytial virus (8) as well as live viral vaccines against Smallpox and Yellow Fever (9). The vast majority of immune perturbations induced by these infections reverted to pre-infection baseline within few weeks after pathogen clearance.

Our approach is also based on comprehensive studies which demonstrate that most immune perturbations in severe COVID-19 recover within a few weeks after symptom onset (10, 11). Bergamaschi et al. (10) showed that the duration and magnitude of immune perturbations is dependent on disease severity and that most immune perturbations resolved within 60 days. Persisting immune perturbations included plasmablasts, naïve B cells, plasmacytoid dendritic cells (pDCs), CD38⁺HLA-DR⁺ activated T cells and innate T cell populations including MAIT cells. However, a more comprehensive analysis of peripheral dendritic cells revealed complete recovery of pDCs in convalescent individuals and is contradictory to the report from Bergamaschi et al. (12). Similarly, a detailed analysis of MAIT cells revealed recovery of MAIT cells in severe COVID-19 within 42-58 days after disease onset (13). Noteworthy, immune perturbations in nasal tissues from hospitalized cases improved during the first 15-61 days of hospitalization and fully

recovered in convalescent individuals within a median of 77 days after symptom onset (14). In conclusion, most immune cell subsets affected by SARS-CoV2 normalize within few weeks after symptom onset.

For our analysis, we used cross-sectional samples spanning a fairly large time window between symptom onset and sample collection for moderate and severe COVID-19 cases (Range of 24-129 days for moderate COVID-19 and 16-184 days for severe COVID-19). We started our analysis with the identification of immune traits (i.e. frequency of subsets or expression levels of functional markers) which were affected by long-term perturbations for two reasons. First, these long-term perturbed immune traits may give clues about the mechanisms regarding the development of Post-Acute Sequelae of COVID-19 (PASC). Second, and importantly for this study, we needed to restrict our analysis to only those traits most likely to be equivalent to pre-COVID baseline, by excluding any long-term perturbed immune traits.

To identify long-term perturbed immune traits, we first compared immune trait values between individuals sampled early (<60 days) and late (>60 days) after symptom onset. We applied linear regression to these data to identify any correlation between immune trait values and time since symptom onset (corrected for age and experimental batch). Long-term affected traits would show a trend towards levels observed in unexposed healthy individuals, even in samples collected months after symptom onset. We observe such patterns mainly in the severe COVID-19 group (Fig. 2). For instance, the frequency of switched B cells is higher in individuals who were sampled early after symptom onset within the severe COVID-19 group (Fig. 2b). In contrast, samples collected later showed frequencies comparable to unexposed healthy individuals. We excluded this immune trait and other traits with similar patterns from the down-stream analysis as the temporal dynamics clearly that early samples are not at baseline.

We thus excluded traits which showed a significant difference in at least one of the two analyses (i.e. Wilcoxon or linear regression) within at least one of the two severity groups (i.e. moderate or severe COVID-19 group) as shown in Figure 1b. While we took this conservative approach to identifying traits that were not at baseline, the vast majority of immune traits in our study neither showed a difference between individuals sampled early and late after symptom onset nor a temporal pattern in the linear regression analysis (Fig. 1b). Furthermore, these traits showed distributions similar to unexposed nominally healthy adults, and were selected for our present analysis.

We cannot rule out the possibility that some immune traits are affected within days and this acute change lasted almost permanently (both of which would to occur to pass our time-regression analysis for inclusion). However, we argue that this “step-change” is unlikely after clearance of an acute viral pathogen. This is also contraindicated by the reports cited above. Notably, some of the long-term perturbed immune traits described in our study overlapped with the long-term perturbations described by Bergamaschi et al. such as naïve B cells and activated CD38⁺HLA-DR⁺ T cells. These cells are from the adaptive immune system and their turn-over and renewal dynamics is usually much longer compared to immune cells of the innate immune arm such as dendritic cells or monocytes. Therefore, it is likely that perturbations are imprinted for a prolonged period in subsets from the adaptive immune arm. We discuss this in the discussion section (Page 18, Line 475).

We clarified this approach in the results section (Page 6, Line 130):

We aimed to identify immune signatures at steady-state which contribute to severe COVID-19. However, cohorts with baseline PBMC samples from patients who had not yet been infected with COVID-19 are not available. In this cross-sectional study, we analyzed PBMC collected after recovery from mild, moderate, severe and critical COVID-19 (Extended Data Figs. 1 and 2), and focused on traits related to the highly significant results from GWAS analysis.

COVID-19 induced immune perturbations can persist after viral clearance and recovery (10, 11). We hypothesized that immune cells may show three potential trajectories following infection. First, a cell trait may not be affected by COVID-19 infection and remain at baseline throughout infection (but still contribute to susceptibility to severe disease). Secondly, a trait may be affected and deviate from unexposed healthy individuals only during active COVID-19 and recover over time after pathogen clearance. Lastly, a trait may change rapidly after viral transmission and remain perturbed after viral clearance without improvement. The latter is likely infrequent, if at all, due to the renewal capacity of the human immune system (15-20). The

second scenario would result in temporal changes of immune traits which would be detectable in samples collected early vs late after symptom onset. To identify traits that might contribute to COVID-19 severity, we took the conservative approach of eliminating traits that showed any evidence of perturbation over time, for which we cannot assert the baseline, pre-COVID-19 value. A caveat to this approach is that a trait which changed from baseline before our earliest measurements, and then remained at that level following our latest measurements, would be included in our analyses. However, we considered that scenario highly unlikely in this dynamic acute infection, particular since viral clearance often occurs early irrespective of disease severity (10) and given the immune system's capacity to recover rapidly from perturbations (15-20).

And page 9, line 208:

We defined stable immune traits as traits which did not show any significant temporal changes in either the moderate or severe COVID-19 group and either the linear regression analysis or the comparison of early and late group (Fig. 1b). As described above, our analysis is conservative in identifying immune traits which change over time.

Furthermore, we added the following text in the discussion to discuss the rationale of our approach in more detail (Page 15, Line 395):

The human immune system rapidly reverts to baseline after pathogen clearance with a composition comparable to pre-infection (1, 21). This rapid immune recovery has been described for several acute viral infections including Ebola (2), Zika (3), Dengue (4, 5), Measles (6), Hantavirus (7) and Respiratory Syncytial Virus (8) as well as live virus vaccines for Smallpox and Yellow Fever (9). In these cases, most immune perturbations recovered within weeks after pathogen clearance. Similarly, most SARS-CoV2-induced immune perturbations normalized within 60-80 days after symptom onset (10, 11, 22). Furthermore, immune perturbations in nasal tissues from severe COVID-19 patients improved already within 15-61 days of hospitalization and normalized after discharge (median of 77 days after symptom onset). Bergamaschi et al. demonstrated that viral titers drop to undetected levels irrespective of disease severity between 13-24 days after symptom onset which overlapped with improvement of immune perturbations (10). Nevertheless, some immune perturbations lasted for months and included naïve B cells, activated CD38⁺HLA-DR⁺ T cells and the expression of CXCR3 (10, 11, 22). Noteworthy, not all the long-term immune perturbations described by Bergamaschi et al. could be replicated in studies with a more detailed focus on individual immune cell subsets including pDCs (12) and MAIT cells (13). In agreement with these studies, we also did not observe any temporal changes of pDC and MAIT cell abundance in individuals recovered from moderate and severe COVID-19. It remains to be determined why there are discrepancies between studies regarding long-term immune perturbations, but reasons could include timing of sample collection and sample cohorts. Overall, based on these studies and our results (Fig. 1b), we argue that samples taken after recovery from COVID-19 mostly reflect the immune system at steady-state and are comparable to pre-infection.

Reviewer #1 (Remarks to the Author):

Liechti and colleagues report on immune phenotypes that predict COVID-19 severity using PBMC samples from 98 patients post-recovery (defined as/assumed steady-state levels) and 173 healthy controls. Mild and moderate patients resolved symptoms by the time of sample collection (medians of 40 and 45 days from onset of symptoms respectively), while patients from the severe and critical groups (77 and 33 days median from onset of symptoms, respectively) showed at least substantial improvement of symptoms.

The authors measure the immune composition of peripheral blood using high-dimensional flow cytometry and they identify distinct chemokine receptor signatures: individuals recovered from severe COVID-19 had increased expression of lung-homing chemokine receptors CCR4, CXCR3, and CX3CR1 on NK cell subsets, and homing-receptor CCR4 on trans. mem. CD4⁺ and CD8⁺ T cells, which can result in exacerbated lung inflammation. In contrast, higher levels of TIGIT were found in mild and moderate group

patients, potentially actively preventing immune pathologies and severe lung damage. In addition, CCR1 and CCR2 were reduced on monocyte subsets in severe cases, potentially associated with a protective role.

The findings of the authors will be of significance to the field and the approach of trying to find predictive markers using post-symptomatic samples is quite unique and inventive (though challenging).

The manuscript is well written, well-structured, and the statistical methods seem sound.

We would like to thank the reviewer for the encouraging feedback. Our response to the reviewer's comments is listed below.

I would like to address the following comments:

Major points:

1a)

The authors seem to have dealt with this to a certain extent by leaving out the critical group in their analysis at page 6 line 130 (inclusion of only moderate and severe group). At page 8 line 190 they take another approach to correct for this effect: they include only the discharged individuals which means that they have included 8 critical patients and excluded 6 severe patients (extended data fig 1). Why do the authors change their approach to correct for this potential bias and what would happen if they compare only moderate and severe groups for this analysis at page 8?

We thank the reviewer for this comment, and we would like to clarify the cohorts and how we used the sample groups in our analysis.

In a first step we aimed to identify any immune trait (frequency of immune cell populations or expression of functional markers), which was affected by long-term perturbations. To this end we focused our analysis on the moderate and severe group because these two groups included cross-sectional samples with the largest range of time between symptom onset and sample collection. This provided increased precision to identify traits which are recovering at a slow rate.

To identify immune traits affected by long-term perturbations we assessed the changes of immune traits over time (time between symptom onset and sample collection). We did this in two ways. First, we used linear regression to identify correlation between immune traits and the time between symptom onset and sample collection. If an immune trait shows a significant correlation between samples collected early and late after symptom onset, then we defined this as a trait in recovery and not yet at baseline in all samples. Examples of such temporal patterns are highlighted in Figure 2b-f. The severe group showed perturbed abundance or expression levels in samples collected early after symptom onset and the samples collected later were closer to the distribution of the 173 unexposed healthy controls.

We used a second approach in parallel which is less sensitive to outliers. In this second approach we split the samples from the moderate and severe COVID-19 groups in two groups, namely short and long recovery group. These samples were grouped based on a cutoff of 60 days between symptom onset and sample collection and we compared these groups using Wilcoxon test. We combined the output from both analyses in Figure 1b.

In our main analysis we aimed to identify immune traits which are predictive for COVID-19 severity. For this, we focused only on immune traits which did not significantly correlate with time of sampling (linear regression analysis) AND did not significantly differ between samples collected early (<60days) and late (>60 days) after symptom onset (Wilcoxon test) in either the moderate or severe group (immune traits in lower left quadrant in Figure 1b). We then used these potentially predictive immune traits and assessed whether they differed in individuals recovered from non-severe (combined mild and moderate group) and

severe (combined severe and critical group) COVID-19 using logistic regression and Benjamini-Hochberg false discovery rate. We first used all samples from all four severity groups (stratified into non-severe and severe groups). However, since some individuals were still hospitalized during sample collection, we argued that these individuals may not be fully recovered from COVID-19 and therefore their immune system might still be perturbed. Therefore, we repeated the analysis and included only samples from individuals which were discharged either at or prior to sample collection. Both analyses gave very similar results (Extended Data Figure 4c and d); thus, symptom improvement despite hospitalization results in normalization of the immune system. Noteworthy, this matches the observations from Roukens et al. which demonstrate that immune perturbations already improve substantially during 15-61 days of hospitalization (14). We highlight a selection of these significant and potentially predictive immune traits in Figures 3-6. For this, we delineated the frequency of all groups individually to see the differences for each individual study group.

Our approach is described in the methods section (Statistical analysis, Page 23, line 607). We added the following paragraph to clarify the logistic regression analysis with all samples compared to only non-hospitalized individuals at sample collection (Page 24, line 644):

We ran the logistic regression analysis with all individuals (N = 98) or only individuals who were not hospitalized or discharged at day of sample collection (N = 71) since hospitalizations at sample collection might point towards persisting symptoms and immune perturbations and affect our analysis (Extended Data Figs. 4c and d).

Fig 2d exemplifies the potential bias that may have resulted from the difference in day of sampling between the groups (severe versus moderate in this case): severe patients at day 100 post onset have higher frequency % than moderate patients at day 100 whereas severe patients at day 40 had lower levels than moderate patients at day 40. Thus, the composition of the groups (sampling date post onset of symptoms) was essential and may have influenced the other analyses (page 6, line 130).

The reviewer understands that this was a factor the authors had no influence on but can the authors comment on this in the manuscript where they have taken this into account in their analyses (for example by comparing only samples taken in a similar time range). At least, the authors can comment in the discussion on how this may have influenced their results.

We thank the reviewer for pointing this out and we agree to some degree with the comment.

It was expected to see more long-term immune perturbations in the severe COVID-19 group since the severity is associated with the magnitude of immune perturbations (23). Therefore, many immune traits might not be affected in individuals with moderate COVID-19 and show levels comparable to the unexposed healthy control group. Noteworthy, in most cases the temporal shifts in the severe group were towards the range of the unexposed healthy control group (Figure 2). Therefore, we are confident that our analysis reflects the correct dynamics of immune trait recovery in long-term perturbations.

We mention this in the discussion (Page 18, Line 479):

Most of these long-term perturbations occurred in severe COVID-19, likely due to increased immune activation (24), and affected mainly B and T cells (Fig 2).

b) In addition, the authors mention they aim to “identify immune perturbations which may contribute to long COVID-19” (line 128 page 6), however it is not very clear if these type of symptomatology is taken into account in their analyses. Potentially, this may even have affected their results, when patients with and without long-COVID-19 symptomatology are considered as one group when studying their immunological profile. Could the authors elaborate on this in the text as well?

We agree with the reviewer that symptoms may have an influence on the immune composition. However, long COVID or Post-Acute Sequelae of COVID-19 (PASC) with lung-associated long-term symptoms seem to have lung-specific immune signatures which are not detectable in blood (25). Only few immune analytes differed in blood between individuals with resolved disease and PASC (26). Furthermore, disease severity during acute stage is not associated with PASC and many factors associated with the risk of PASC are only

detectable early during acute infection (27). Su et al. demonstrated that the composition of most immune cell subsets does not differ between recovered COVID-19 patients and PASC, and the few immune traits associated with PASC depend on the type of symptoms (27). Therefore, the data suggests that immune signatures associated with PASC occur locally in tissues and that the risk of PASC is dependent on other factors such as Epstein-Barr virus viremia, auto-antibodies and bystander activation of CMV-specific CD8+ T cells (27). In agreement, a recent study showed that while PASC patients had higher frequencies of SARS-CoV2-specific T cells, the phenotype of total T cells did not differ compared to individuals with resolved COVID-19 (28).

Data regarding persisting symptoms was not collected in our cohorts. However, the results described above suggest that the immune composition in blood is minimally affected in PASC. With our analysis in Figures 1b and 2 we aimed to describe immune traits which recover slowly and therefore might contribute to or be affected by PASC. We discussed our analysis approach in detail above.

For subsequent analysis in Figures 3-6 we only used immune traits which did not differ between individuals sampled early and late after symptom onset. Therefore, these immune traits rapidly recovered after symptom onset if they were affected by COVID-19 and represent baseline levels. Thus, these traits are unlikely to contribute to persisting symptoms and PASC.

We added a paragraph in the discussion to highlight this potential concern (Page 18, Line 490):

Post-acute Sequelae of COVID-19 (PASC) occurs in a fraction of individuals resulting in persisting symptoms, but its occurrence did not correlate with disease severity (27). Several factors early during acute infection are associated with the risk of developing PASC such as Epstein-Barr virus (EBV) viremia, auto-antibodies, and bystander activation of Cytomegalovirus (CMV)-specific T cells (27). In contrast, the immune composition in the blood remained largely unaffected (27). Noteworthy, immune signatures associated with PASC occurred in lungs but not in blood (25). In our study, we can not delineate immune trajectories associated with PASC since such information was not collected in our cohorts. However, based on these studies the impact of PASC on the immune composition seems to be minor in recovered individuals.

c) in the abstract the authors write: line 29 “focused only on those immune signatures returning to steady-state.” This may need some clarification, since different approaches are followed in different analyses in different paragraphs of the result and it does not seem the case that the authors have included only parameters that returned to steady-state levels in all groups. Please clarify.

We outlined our procedure under subsection 1a and hope that our more detailed description will clarify the confusion.

2)

The findings on the naïve CD4+ T cells and naïve CD8 cells (fig 2c and 3c) are significant and of interest to the field. Can the authors elaborate on those findings in the discussion and describe how this relates to the findings of others published in literature to date?

We thank the reviewer for the comment, and we agree that these observations are interesting. Co-expression of CD38 and HLA-DR is often used to define activated T cells (29, 30). In general, naïve T cells constitutively express CD38 (31). In untreated chronic HIV-1 infection, CD38 is reduced within CD45RA⁺CD4⁺ T cells (29). However, in that report there is no distinction between naïve T cells and effector memory T cells, which both express CD45RA. Thus, the functional properties of CD38^{low/-} naïve CD4 and CD8 T cells remain unknown.

We added the following sentence to the result section in more detail (Page 8, Line 182):

CD38 on naïve T cells is down-regulated in HIV-1 but their functional properties remain unclear (29, 31).

Minor points:

3)

With regards to the controls, could the authors describe the vaccination status or year of inclusion? Can vaccination prior to sampling have affected the data?

We agree that this information is important. Our samples were collected prior to availability of SARS-CoV2 vaccines. We added the following sentence to the methods section (Page 20, Line 529):

All samples were collected prior to the availability of vaccines against SARS-CoV2. Furthermore, samples from the healthy control group were collected prior to the emergence of SARS-CoV2 and thus represent unexposed healthy individuals.

We also specify throughout the text and in figure captions that the healthy control group represent unexposed individuals.

4)

In the top paragraph at page 7 the authors describe “the expression levels over time”. Can the authors for clarity write there and/or in the legend that each patient is sampled only once? Just to distinguish from temporal dynamics analysis within patients.

We agree that we need to clarify that none of these samples represent longitudinal sampling from the same individual and that samples are purely cross-sectional. We added the following sentence to the results section to clarify this (Page 6, Line 132):

In this cross-sectional study, we analyzed PBMC collected after recovery from mild, moderate, severe and critical COVID-19 (Extended Data Figs. 1 and 2) and focused on traits related to the highly significant results from GWAS analysis.

We also added the following text to the method section (Page 20, Line 529):

In this cross-sectional study, one sample per individual was measured.

Reviewer #2 (Remarks to the Author):

The authors describe differences in steady state immune phenotypes after mild and severe COVID-19. The methods and results are well described, but the conclusion drawn is not sound.

The large number in each group provides confidence in the results and will allow future studies to understand both recovery and potentially also an association with long COVID. The findings of note include reduced levels of chemokine receptors on monocyte subsets and reduced frequencies of type I IFN-producing pDCs, demonstrating differences in cell types that play a role in the earliest host response to infection.

We would like to thank the reviewer for the encouraging feedback. Our response to the reviewer's comments is listed below.

The title, abstract and text that refer to the results “predicting” COVID19 severity should be modified to reflect that the results are associated with severe COVID19. Alternatively, data to demonstrate the phenotypes associated with severe COVID19 are indeed present prior to infection, could be added. Perhaps there is information regarding COVID19 infection for some of the comparators (healthy individuals) that would provide a source of these data.

We discussed our approach extensively above in the section where we reply to concerns, which were raised by all three reviewers. We agree that a study with pre-infection samples would be more robust to find predictive immune signatures for severe COVID-19. Unfortunately, such cohorts are not available to the best of our knowledge. However, we are convinced that our approach is valid based on the arguments outlined above and our results will serve as a guide for studies with larger cohorts.

Our cohort of 173 healthy individuals was sampled prior to COVID-19 and therefore this control group is not affected by COVID-19 related immune changes. Thus, our control group serves as a true representation of COVID-19 unexposed individuals. However, we do not have any information regarding COVID-19 infection or severity from this cohort.

As written, the manuscript describes traits at “baseline” (line 400/401) for data used to generate Figure 1b from analyzing differences explained as “before and after 60 days of symptom onset” on line 909; the methods describe these as “convalescent” and state that the PBMCs from those in severe and critical cases were from individuals who were recovered (line 432). Plots shown in Figure 2 imply that the earliest time points were 2-3 weeks after symptom onset. In looking at this information again, I realize that I’ve misunderstood that samples were collected from the same individual at different times post infection. The methods as well as text should plainly state that single samples from individuals who had COVID19 symptoms of various degrees were used in the analysis of changes over time, or provide the specific consecutive days post-illness onset together with the change over time.

Reviewer #1 made a similar suggestion and now we added the following sentence to the results section to clarify this (Page 6, Line 132):

In this cross-sectional study, we analyzed PBMC collected after recovery from mild, moderate, severe and critical COVID-19 (Extended Data Figs. 1 and 2) and focused on traits related to the highly significant results from GWAS analysis.

We also added the following text to the method section (Page 20, Line 529):

In this cross-sectional study, one sample per individual was measured.

Some of the data presented may mislead readers who are looking at images only, as it appears different cell numbers are presented in the figures; an example is Figure 3d; while the % of cells expressing TIGIT is clearly stated, it would help to either present the same number of cells in the plots or to state the number of cells plotted, in the figure legend.

We thank the reviewer for this comment. We kindly disagree with this suggestion. In flow cytometry, data is often represented without normalized cell counts. As the reviewer pointed out, we were interested in abundance and clearly state the frequency of TIGIT⁺ cells in the plots which is in accordance with the journal’s guidelines and according to best practices in the flow cytometry community.

Reviewer #3 (Remarks to the Author):

This manuscript by Liechti et al. aimed to determine the unique immune signatures that predispose to severe COVID-19. They assessed peripheral blood immune cells and cytokine receptor expression profiles in subjects recovered from mild, moderate, severe, and critical COVID-19 diseases. They proposed that measuring immune composition in recovered subjects when immune responses restore back to baseline would provide insight in the key and differential factors that drive severe diseases. By using 28-color high dimensional flow cytometry, they found that those who recovered from severe COVID-19 had elevated expressions of lung-homing receptors CCR4, CXCR3 and CV3CR1 on NK cells, and also had increased CCR4 expression on memory CD4 and CD8 T cells. CD8 T cells (central memory and

effector T cells) also showed reduced levels of TIGIT, which had been shown to dampen disease pathogenesis caused by influenza. The number of central memory CD4+ and CD8+ T cells were reduced and CD8+ effector and terminal memory T cells were increased in those recovered from severe disease. MAIT cells were also reduced in frequency but with increased TIGIT expression.

Monocytes also displayed decreased CCR1 and CCR2 expressions that have been reported to dampen SARS-CoV-2-induced inflammation in vivo. These results indicate excessive homing of NK and T cells into the lung, possibly driving excessive inflammation and tissue damage. Plasmacytoid dendritic cells (pDCs) are the primary source of antiviral cytokines type I interferons (IFNs), but the authors show pDC numbers were lower in those who recovered from severe COVID-19. IFN receptor (IFNAR2) expression was increased on pDCs and monocytes from those recovered from severe COVID-19, possibly indicating reduced antiviral responses due to reduced productions of type I IFNs by pDCs despite increased IFNAR2 expression. The authors concluded that these immune features after recovery from severe COVID-19 reflect the immune system at steady-state and are comparable to pre-infection and these immune signatures predict the trajectory of disease severity.

This study provides novel immunological insights in COVID-19, especially the NK homing, TIGIT expressions on T/MAIT cells as well as IFNAR2 expressions, and while the flow cytometry is mostly methodologically sound, there are key questions on donor characteristics, vaccine status and other potential confounding factors that need to be included to solidify authors claims

We would like to thank the reviewer for the encouraging feedback. Our response to the reviewer's comments are listed below.

General comments

Would the existing medical conditions such as obesity, asthma, or COPD impact the identified immune traits identified in this study ? were these info recorded ?

Unfortunately, we only have BMI information for the severe and critical COVID-19 group and no other information regarding medical preconditions were available.

Were vaccination status recorded for all recruited participants and how would this impact this results and conclusions of this study.

Reviewer #1 raised a similar concern. All samples were collected before a vaccine was available. We added the following sentence to the method section (Page 20, Line 529) to highlight this:

All samples were collected prior to the availability of vaccines against SARS-CoV2. Furthermore, samples from the healthy control group were collected prior to the emergence of SARS-CoV2 and thus represent unexposed healthy individuals.

The results section particularly when describing cell phenotypes and their surface receptors, it is important to also describe what these receptors do in the context of cell recruitment/activation and in infection/immunity for better clarity to the readers.

Please discuss in the discussion section on what do the reduced or increased assessed immune cells and their activation or receptor expression (that were assessed in this study) mean for the immune responses and disease severity. Ie What does reduced MAIT and other cells mean in terms of immune responses? More inflammation and less tissue repair ?

We thank the reviewer for the comment and we agree that more information is helpful to better understand the implications of our results. However, our comprehensive immune profiling approach includes numerous immune cell subsets and molecules. Thus, it is not feasible to include detailed information for all of these immune traits. We added this information throughout the manuscript where we think it is most critical but had to keep the word count limitations in mind.

CCR3 (Page 8, Line 190):

Basophils regulate T_H2 immune responses (32) and express CCR3 which induces cell migration and release of histamine and leukotriene (33).

CD95 (Page 9, Line 200):

Fas (CD95) induces cell death and is important for immune homeostasis (34) and is expressed on memory and effector T cells (31).

Chemokine receptors - Introduction (Page 3, Line 66)

The pro-inflammatory chemokine receptors CCR1, CCR2, CXCR3 and CX3CR1 are expressed on myeloid cells (35, 36), T cells (37, 38) and NK cells (39-42). These chemokine receptors orchestrate immune responses in the lungs including allergies (43), M. tuberculosis (44) and Influenza (42). In contrast, CCR4 (45, 46), CCR5 (47, 48) and CXCR6 (49) are mainly expressed on conventional and innate T cells and regulate cell influx to the lung and mucosal tissues (47-49). CCR4 is critical for homing of immune cells to skin (45, 46), while CCR9 regulates cell migration to mucosal tissues (46, 50). Thus, chemokine receptors can induce non-redundant, tissue-specific cell migration. Severe COVID-19 has been associated with perturbations in chemokine levels as well as expression of chemokine receptors highlighting their importance in immunopathology (23, 51).

TIGIT (Page 10, Line 257)

The co-inhibitory receptor TIGIT limits T cell activation and viral immunopathology (52).

MAIT cells (Page 11, Line 265)

MAIT cells recognize unique microbial riboflavin-like antigens (53), respond rapidly through the expression of cytokines and cytotoxic granzyme B (54) and have various functions including pathogen clearance, tissue repair but also immunopathology and inflammation (53).

pDCs (Page 4, line 101)

In addition, our data revealed reduced levels of plasmacytoid DCs (pDCs), which produce high levels of type I interferon (37).

Pre-DCs (Page 14, Line 358).

The latter are precursors and differentiate into conventional cDC1 and cDC2 (35)

CD14⁻ and CD14⁺ DC3s (page 11, Line 290)

CD14⁻ and CD14⁺ DC3s are inflammatory cells and more efficient in activating and polarizing T helper responses compared to conventional cDC2s (35).

Specific comments

Line 99, PBMC from 173 healthy individuals... it is more appropriate to indicate that they are recovered individuals.

We thank the reviewer for the comment. Blood samples from these 173 individuals were collected prior to the COVID-19 pandemic. Therefore, this control group represents truly SARS-CoV2 unexposed individuals. We highlight this now in the methods section (Page 20, Line 529):

All samples were collected prior to the availability of vaccines against SARS-CoV2. Furthermore, samples from the healthy control group were collected prior to the emergence of SARS-CoV2 and thus represent unexposed healthy individuals.

Line 100, please include donor demographics and characteristics.

We thank the reviewer for his comment. The demographic information is listed in Extended Data Figure 2.

Line 104, please indicate that these 4 unique sets of markers are.

The list of markers for each panel can be found in Supplementary Table 1 and the list of reagents is added as Supplementary Table 2. We now refer to these tables in the text.

Extended data fig 1a and line 132, what does the y-axis represent? and the numbers below the box of the boxplot ? median day of symptom onset and sample collection ? why is this shorter for critical conditions requiring ventilation vs other groups?

We thank the reviewer for the comment. We agree that Extended Data Figure 1a is not sufficiently intuitive.

We clarify this now in the figure caption for Extended Data Figure 1a:

Distribution of days between symptom onset and sample collection (x-axis) is shown for each COVID-19 severity group (y-axis; mild, moderate, severe and critical COVID-19) as boxplot. Individual datapoints are highlighted as black dots. In addition, histograms (y-axis) depict the distribution of samples for each COVID-19 severity group. The number below each boxplot shows median days between symptom onset and sample collection per group. Red dashed line indicates 60 days cutoff which was used for analysis shown in Figures 1b and 2a and Extended Data Figure 5a. Information about time between symptom onset and sample collection was unavailable for two samples from the mild COVID-19 group.

The cohort for the severe and critical COVID-19 group was established with the goal to collect samples longitudinally at enrollment and approximately at 3, 7, 14, 28, 84 and 180 days after enrollment. For the critical group, samples could be collected while patients remained in the hospital until these patients recovered substantially and were discharged. However, these individuals often did not follow up after their discharge from hospitalization which is why samples from 84 and 180 days after enrollment are mostly missing for these individuals. This resulted in shorter median time between symptom onset and sampling for the critical COVID-19 group.

Throughout the manuscript it is stated that PBMCs were obtained from recovered individuals, but for Fig 2 b-f there are data presented as “healthy”, were these data compared to recovered subjects and different disease severity?

These 173 individuals were enrolled as part of the Vaccine Research Center cohort and samples were collected prior to the COVID-19 pandemic. Therefore, this cohort reflects truly SARS-CoV2 unexposed individuals. We use the data from this control group throughout the manuscript (referred to as healthy)

We added the following sentence to the methods section (Page 20, Line 529):

All samples were collected prior to the availability of vaccines against SARS-CoV2. Furthermore, samples from the healthy control group were collected prior to the emergence of SARS-CoV2 and thus represent unexposed healthy individuals.

Line 153 – 156, need to state what CD38+ and CD38- represents ? T cell activation ? it should also include that it was the “early” decrease in CD38+ and decrease in CD38- naïve T cells that were associated with severe diseases.

Reviewer #1 raised a similar question. We added the following sentence to the results section to discuss in more detail (Page 8, Line 182):

CD38 on naïve T cells is down-regulated in HIV-1 but their functional properties remain unclear (29, 31).

Line 163, given basophil CCR3 expression and IFNAR2 (line 382) was altered in those recovered from severe COVID-19, was basophil frequency also altered ?

We thank the reviewer for the comment. We did not observe significant differences in the frequency of basophils between the COVID-19 severity groups. All immune traits which significantly differed between non-severe (mild and moderate groups combined) and severe (severe and critical groups combined) COVID-19 are listed in Extended Data Figure 3b if all individuals are included in the analysis and in Extended Figure 4b if only non-hospitalized individuals at time of sample collection were included in the analysis. We clarify this in the results section (Page 9, Line 219 and Page 9, Line 225):

All significant results from this analysis are listed in Extended Data Fig. 3b.

All significant results from this analysis are listed in Extended Data Fig. 4b.

Line 188, patients with severe and critical disease were still hospitalized at the time of sample collection, and their data was excluded and those who were discharged were included in the data analyses. Would it be interesting to look at whether the assessed parameters are also altered in this group vs other groups th

We appreciate the comment from the reviewer and agree that it would be interesting to know if differences in the immune composition exists between non-hospitalized and hospitalized individuals at sample collection. As described above, we assigned individuals sampled at the day of discharge to the non-hospitalized group at sample collection. We ran logistic analysis and looked for differences between hospitalized and non-hospitalized patients at sample collection for the severe and critical COVID-19 group and both groups combined. We corrected for age and experiment batch and used Benjamini-Hochberg false discovery rate to correct for multiple testing. We used the 1365 immune traits after excluding immune traits which showed temporal changes (Fig. 1b). The graph below shows the volcano plots with $-\log_{10}(p\text{-value})$ on y-axis and fold change (x-axis). None of the traits differed significantly between non-hospitalized and hospitalized individuals regardless of severity group. Therefore, we decided to show this analysis to the reviewer but decided not to include it in the manuscript.

EDITORIAL NOTE: FIGURE REDACTED AT AUTHOR'S REQUEST.

Line 196, non-hospitalized patients refer to discharged or those requiring no hospitalization ?

This refers to individuals who were discharged prior or at day of sample collection. Some required hospitalization but were discharged prior to sample collection. This info can be found in Extended Data Figure 1e.

We added the following sentence in Page 10, Line 232 to clarify this:

Non-hospitalized refers to individuals who did not require hospitalization or were discharged from the hospital prior to or at day of sample collection. Information about hospitalization can be found in Extended Data Figs. 1e and 2.

Line 194, “65 significant hits were shared between...” and “only 6 new immune traits were discovered ...”

Please describe what those hits are and in addition to the venn diagram are these hits presented in the figs at all ?

We added this info in the Source data for Extended Data Figure 4. For the final submission, we will submit all source data for the main and Extended Data Figures. In addition, the flow cytometry raw data will be uploaded on FlowRepository once the manuscript gets accepted.

Line 249, “the chemokine receptor profile on dendritic cells did not differ substantially between...” please indicate the fig this is referred to.

We refer now to Extended Data Figure 3 and 4.

Line 263 – 264, “IFNAR2 expression was lowest on naïve B cells and highest on pDCs and cDC1s...”, and in the next sentence “we found increased expression of IFNAR2 on monocyte and dendritic cell subsets, except for cDC1s and pDCs.” These two sentences conflict with each other.

Was IFNAR2 expression tracked as in Fig2e on these cells?

We thank the reviewer for this comment. We agree that these two sentences are misleading. We changed these sentences accordingly (Page 12, Line 307):

*IFNAR2 expression was lowest on naïve B cells and highest on pDCs and cDC1s within the 173 unexposed healthy individuals (Fig. 1a). Noteworthy, IFNAR2 expression could be detected on most subsets including cDC2s, DC3s and monocyte subsets (Fig. 1a). Next, we assessed how the expression of IFNAR2 on these immune cell subsets differed between the unexposed healthy controls and the different COVID-19 severity groups (Figs. 4d and e). Monocytes and dendritic cells, except cDC1s and pDCs, showed elevated expression of IFNAR2 in individuals recovered from severe and critical COVID-19 compared to unexposed healthy controls and mild and moderate COVID-19 group (Bonferroni-adjusted P-value range 0.04 - 4.95*10⁻⁵) (Figs. 4d and e).*

We defined IFNAR2 expression on all myeloid and B cells subsets as shown in Figure 1b. We did not measure IFNAR2 on T cells and NK cells as the flow cytometry staining panels differed between DCs/B cells and T/NK cells (Supplementary table 1). Our staining panels and analysis are outlined in the method section. The IFNAR2 measurements which did not differ between samples collected early and late after symptom onset (i.e. are not affected by long-term perturbations) within the moderate and severe group (as shown in Fig. 1b) were used in the logistic regression analysis. There we compared immune traits from these individuals recovered from non-severe (mild/moderate) with severe (severe/critical) COVID-19. If these IFNAR2 traits were significant in the logistic regression analysis, then we depicted them in Extended Data Figs. 3 and 4 which lists all significant results from the logistic regression analysis.

Fig 4d, the line graph beside the flow histogram, as the data is not continuous, perhaps this is best presented as bar graph.

We aimed to show how individual immune traits change with severity and think that a connected line is most suitable to visualize these dynamics. We do not think that the x-axis will be misleadingly interpreted as a temporal shift.

Line 280, would be interesting to look at whether the differentially expressed immune signatures assessed are also altered in the immune perturbed group that was excluded from the study and whether these altered immune phenotypes also contribute to the disease severity.

We did not exclude any patients except when we did the logistic regression analysis with only those who were not hospitalized at day of sample collection (Extended Data Fig. 4). We only excluded long-term perturbed immune traits as these do not reflect baseline immune composition.

Line 289, “from each chemokine receptor panels... 5 and 6 significant clusters resemble innate-like T cells” please indicate where they are in fig 5b.

The goal of Figure 5b is to get a better overview of the magnitude of difference (fold change) between individuals recovered from non-severe and severe COVID-19. We only highlight the cluster numbers in Figure 5b for the clusters which significantly differed between individuals recovered from non-severe and severe COVID-19. That being said, the same cluster numbers are used for the heatmaps in Supplementary Figures 8-13 to highlight their expression profile of immune receptors. In Figure 5c we further focused on the expression profile of innate T cell populations. The cluster numbers are highlighted on the right of the small heatmaps. Therefore, all clusters can be tracked across the manuscript. We mention these cluster number in the text (Page 13, Line 338):

From each chemokine receptor panel (CR1 and CR2) 5 and 6 significant clusters ($pFDR < 0.01$) resembled innate-like T cells, respectively. Four clusters (clusters 34, 35, 37 and 38) from CR1 and one (cluster 3) from CR2 panel were MAIT cells as defined by T cell receptor (TCR) $V\alpha 7.2$ and CD161 (Fig. 5c).

Line 297 and 298, should the fig referred in the text (Fig. 5d) be Fig 5c?

We thank the reviewer for the comment. This is indeed a mistake, and we refer the sentence now to Fig. 5c.

Please indicate what type of T cells $V\delta 2V\gamma 9$ refers to.

We add this information to the results section (Page 13, Line 347).

$V\gamma 9V\delta 2$ T cells recognize unique microbial phospho-antigens and respond rapidly through secretion of cytokines or cytotoxic molecules and thus may exacerbate inflammation (55). Noteworthy, we observed a shift from $CCR7^+CD27^+$ naïve-like $V\delta 2V\gamma 9$ T cells (CR2 cluster 30) towards effector-like cells (CR2 clusters 20, 24 and 25) which lack CCR7 and CD27 in individuals recovered from severe and critical COVID-19 (Fig. 5d) (56).

Are NK cell numbers increased or decreased in different disease states?

We did not see any significant differences between individuals recovered from non-severe and severe COVID-19 beyond the immune traits listed in Extended Data Figure 3b (if all individuals are included in

logistic regression analysis) and 4b (only non-hospitalized individuals at day of sample collection were included in logistic regression analysis). We did not see any significant differences in NK cell abundance. Maucourant et al. described reduced NK cell abundance in active COVID-19 (57). However, we measured samples after recovery and based on our analysis, these affected NK cell subsets already fully recovered in our samples irrespective of disease severity.

Reductions in cell frequency in peripheral blood such as cDC1s early in days after symptom onsets, could this be due to infiltration of cDC1s into the lung (thus a reduction in blood)? And therefore indicate early antigen presentation? why weren't all cell types and surface receptors tracked through days after symptom onsets (like in Fig2 for B, cDCs, and basophils)? ie CD8 effector T and MAIT cells, monocytes?

As pointed out above, our analysis pipeline consisted of several steps where we filtered out immune traits (i.e. frequency of subsets, frequency of subset expressing a functional marker and the expression levels of functional marker as mean fluorescence intensity):

1. We assessed whether expression of functional markers occurred as described in Page 6, Line 121. From this step we continued with 1758 immune traits out of a total of 3787 original traits.
2. In the next step we identified long-term perturbed immune traits as shown in Fig. 1b and described above and in the manuscript. From this analysis, 1365 immune traits did not show any long-term perturbations.
3. Finally, we wanted to assess if there are any immune traits at baseline which differed between non-severe (mild and moderate samples combined) and severe (severe and critical samples combined) COVID-19 group. We only used the immune traits at baseline (N = 1365) as described in point 2.

Therefore, we did measure all receptors on all immune cell subsets or the abundance of all immune cell subsets in all individuals but based on our analysis only about 1/3 of the original 3787 immune traits remained for assessing immune traits which are predictive for COVID-19 severity.

Literature

1. P. Marrack, J. Scott-Browne, M. K. MacLeod, Terminating the immune response. *Immunol Rev* **236**, 5-10 (2010).
2. A. K. McElroy, R. S. Akondy, C. W. Davis, A. H. Ellebedy, A. K. Mehta, C. S. Kraft, G. M. Lyon, B. S. Ribner, J. Varkey, J. Sidney, A. Sette, S. Campbell, U. Stroher, I. Damon, S. T. Nichol, C. F. Spiropoulou, R. Ahmed, Human Ebola virus infection results in substantial immune activation. *Proc Natl Acad Sci U S A* **112**, 4719-4724 (2015).
3. P. Tonnerre, J. G. Melgaco, A. Torres-Cornejo, M. A. Pinto, C. Yue, J. Blumel, P. S. F. de Sousa, V. D. M. de Mello, J. Moran, A. M. B. de Filippis, D. Wolski, A. Grifoni, A. Sette, D. H. Barouch, R. C. Hoogeveen, S. A. Baylis, G. M. Lauer, L. L. Lewis-Ximenez, Evolution of the innate and adaptive immune response in women with acute Zika virus infection. *Nat Microbiol* **5**, 76-83 (2020).
4. U. K. Nivarthi, H. A. Tu, M. J. Delacruz, J. Swanstrom, B. Patel, A. P. Durbin, S. S. Whitehead, K. K. Pierce, B. D. Kirkpatrick, R. S. Baric, N. Nguyen, D. E. Emerling, A. M. de Silva, S. A. Diehl, Longitudinal analysis of acute and convalescent B cell responses in a human primary dengue serotype 2 infection model. *EBioMedicine* **41**, 465-478 (2019).
5. A. Rouers, M. H. Y. Chng, B. Lee, M. P. Rajapakse, K. Kaur, Y. X. Toh, D. Sathiakumar, T. Loy, T. L. Thein, V. W. X. Lim, A. Singhal, T. W. Yeo, Y. S. Leo, K. A. Vora, D. Casimiro, B. Lim, L. Tucker-Kellogg, L. Rivino, E. W. Newell, K. Fink, Immune cell phenotypes associated with disease severity and long-term neutralizing antibody titers after natural dengue virus infection. *Cell Rep Med* **2**, 100278 (2021).
6. B. M. Laksono, R. D. de Vries, R. J. Verburgh, E. G. Visser, A. de Jong, P. L. A. Fraaij, W. L. M. Ruijs, D. F. Nieuwenhuijse, H. J. van den Ham, M. P. G. Koopmans, M. C. van Zelm, A. Osterhaus, R. L. de Swart, Studies into the mechanism of measles-associated immune suppression during a measles outbreak in the Netherlands. *Nat Commun* **9**, 4944 (2018).
7. T. Lindgren, C. Ahlm, N. Mohamed, M. Evander, H. G. Ljunggren, N. K. Bjorkstrom, Longitudinal analysis of the human T cell response during acute hantavirus infection. *J Virol* **85**, 10252-10260 (2011).
8. A. Jozwik, M. S. Habibi, A. Paras, J. Zhu, A. Guvenel, J. Dhariwal, M. Almond, E. H. C. Wong, A. Sykes, M. Maybeno, J. Del Rosario, M. B. Trujillo-Torralbo, P. Mallia, J. Sidney, B. Peters, O. M. Kon, A. Sette, S. L. Johnston, P. J. Openshaw, C. Chiu, RSV-specific airway resident memory CD8+ T cells and differential disease severity after experimental human infection. *Nat Commun* **6**, 10224 (2015).
9. J. D. Miller, R. G. van der Most, R. S. Akondy, J. T. Glidewell, S. Albott, D. Masopust, K. Murali-Krishna, P. L. Mahar, S. Edupuganti, S. Lalor, S. Germon, C. Del Rio, M. J. Mulligan, S. I. Staprans, J. D. Altman, M. B. Feinberg, R. Ahmed, Human effector and memory CD8+ T cell responses to smallpox and yellow fever vaccines. *Immunity* **28**, 710-722 (2008).
10. L. Bergamaschi, F. Mescia, L. Turner, A. L. Hanson, P. Kotagiri, B. J. Dunmore, H. Ruffieux, A. De Sa, O. Huhn, M. D. Morgan, P. P. Gerber, M. R. Wills, S. Baker, F. J. Calero-Nieto, R. Doffinger, G. Dougan, A. Elmer, I. G. Goodfellow, R. K. Gupta, M. Hosmillo, K. Hunter, N. Kingston, P. J. Lehner, N. J. Matheson, J. K. Nicholson, A. M. Petrunkina, S. Richardson, C. Saunders, J. E. D. Thaventhiran, E. J. M. Toonen, M. P. Weekes, I. Cambridge Institute of

- Therapeutic, C. B. C. Infectious Disease-National Institute of Health Research, B. Gottgens, M. Toshner, C. Hess, J. R. Bradley, P. A. Lyons, K. G. C. Smith, Longitudinal analysis reveals that delayed bystander CD8+ T cell activation and early immune pathology distinguish severe COVID-19 from mild disease. *Immunity* **54**, 1257-1275 e1258 (2021).
11. F. J. Ryan, C. M. Hope, M. G. Masavuli, M. A. Lynn, Z. A. Mekonnen, A. E. L. Yeow, P. Garcia-Valtanen, Z. Al-Delfi, J. Gummow, C. Ferguson, S. O'Connor, B. A. J. Reddi, P. Hissaria, D. Shaw, C. Kok-Lim, J. M. Gleadle, M. R. Beard, S. C. Barry, B. Grubor-Bauk, D. J. Lynn, Long-term perturbation of the peripheral immune system months after SARS-CoV-2 infection. *BMC Med* **20**, 26 (2022).
 12. E. Winheim, L. Rinke, K. Lutz, A. Reischer, A. Leutbecher, L. Wolfram, L. Rausch, J. Kranich, P. R. Wratil, J. E. Huber, D. Baumjohann, S. Rothenfusser, B. Schubert, A. Hilgendorff, J. C. Hellmuth, C. Scherer, M. Muenchhoff, M. von Bergwelt-Baildon, K. Stark, T. Straub, T. Brocker, O. T. Keppler, M. Subklewe, A. B. Krug, Impaired function and delayed regeneration of dendritic cells in COVID-19. *PLoS Pathog* **17**, e1009742 (2021).
 13. T. Parrot, J. B. Gorin, A. Ponzetta, K. T. Maleki, T. Kammann, J. Emgard, A. Perez-Potti, T. Sekine, O. Rivera-Ballesteros, C.-S. G. Karolinska, S. Gredmark-Russ, O. Rooyackers, E. Folkesson, L. I. Eriksson, A. Norrby-Teglund, H. G. Ljunggren, N. K. Bjorkstrom, S. Aleman, M. Buggert, J. Klingstrom, K. Stralin, J. K. Sandberg, MAIT cell activation and dynamics associated with COVID-19 disease severity. *Sci Immunol* **5**, (2020).
 14. A. H. E. Roukens, C. R. Pothast, M. Konig, W. Huisman, T. Dalebout, T. Tak, S. Azimi, Y. Kruize, R. S. Hagedoorn, M. Zlei, F. J. T. Staal, F. J. de Bie, J. J. M. van Dongen, S. M. Arbous, J. L. H. Zhang, M. Verheij, C. Prins, A. M. van der Does, P. S. Hiemstra, J. J. C. de Vries, J. J. Janse, M. Roestenberg, S. K. Myeni, M. Kikkert, M. Yazdanbakhsh, M. H. M. Heemskerck, H. H. Smits, S. P. Jochems, B.-C. g. in collaboration with, C.-L. g. in collaboration with, Prolonged activation of nasal immune cell populations and development of tissue-resident SARS-CoV-2-specific CD8(+) T cell responses following COVID-19. *Nat Immunol* **23**, 23-32 (2022).
 15. R. Sender, R. Milo, The distribution of cellular turnover in the human body. *Nat Med* **27**, 45-48 (2021).
 16. A. A. Patel, Y. Zhang, J. N. Fullerton, L. Boelen, A. Rongvaux, A. A. Maini, V. Bigley, R. A. Flavell, D. W. Gilroy, B. Asquith, D. Macallan, S. Yona, The fate and lifespan of human monocyte subsets in steady state and systemic inflammation. *J Exp Med* **214**, 1913-1923 (2017).
 17. A. A. Patel, F. Ginhoux, S. Yona, Monocytes, macrophages, dendritic cells and neutrophils: an update on lifespan kinetics in health and disease. *Immunology* **163**, 250-261 (2021).
 18. K. Liu, C. Waskow, X. Liu, K. Yao, J. Hoh, M. Nussenzweig, Origin of dendritic cells in peripheral lymphoid organs of mice. *Nat Immunol* **8**, 578-583 (2007).
 19. L. Westera, J. Drylewicz, I. den Braber, T. Mugwagwa, I. van der Maas, L. Kwast, T. Volman, E. H. van de Weg-Schrijver, I. Bartha, G. Spierenburg, K. Gaiser, M. T. Ackermans, B. Asquith, R. J. de Boer, K. Tesselaar, J. A. Borghans, Closing the gap

- between T-cell life span estimates from stable isotope-labeling studies in mice and humans. *Blood* **122**, 2205-2212 (2013).
20. J. E. Mold, P. Reu, A. Olin, S. Bernard, J. Michaelsson, S. Rane, A. Yates, A. Khosravi, M. Salehpour, G. Possnert, P. Brodin, J. Frisen, Cell generation dynamics underlying naive T-cell homeostasis in adult humans. *PLoS Biol* **17**, e3000383 (2019).
 21. P. Brodin, M. M. Davis, Human immune system variation. *Nat Rev Immunol* **17**, 21-29 (2017).
 22. H. A. Shuwa, T. N. Shaw, S. B. Knight, K. Wemyss, F. A. McClure, L. Pearmain, I. Prise, C. Jagger, D. J. Morgan, S. Khan, O. Brand, E. R. Mann, A. Ustianowski, N. D. Bakerly, P. Dark, C. E. Brightling, S. Brij, Circo, T. Felton, A. Simpson, J. R. Grainger, T. Hussell, J. E. Konkel, M. Menon, Alterations in T and B cell function persist in convalescent COVID-19 patients. *Med (N Y)* **2**, 720-735 e724 (2021).
 23. D. Mathew, J. R. Giles, A. E. Baxter, D. A. Oldridge, A. R. Greenplate, J. E. Wu, C. Alanio, L. Kuri-Cervantes, M. B. Pampena, K. D'Andrea, S. Manne, Z. Chen, Y. J. Huang, J. P. Reilly, A. R. Weisman, C. A. G. Ittner, O. Kuthuru, J. Dougherty, K. Nzingha, N. Han, J. Kim, A. Pattekar, E. C. Goodwin, E. M. Anderson, M. E. Weirick, S. Gouma, C. P. Arevalo, M. J. Bolton, F. Chen, S. F. Lacey, H. Ramage, S. Cherry, S. E. Hensley, S. A. Apostolidis, A. C. Huang, L. A. Vella, U. P. C. P. Unit, M. R. Betts, N. J. Meyer, E. J. Wherry, Deep immune profiling of COVID-19 patients reveals distinct immunotypes with therapeutic implications. *Science* **369**, (2020).
 24. C. Lucas, P. Wong, J. Klein, T. B. R. Castro, J. Silva, M. Sundaram, M. K. Ellingson, T. Mao, J. E. Oh, B. Israelow, T. Takahashi, M. Tokuyama, P. Lu, A. Venkataraman, A. Park, S. Mohanty, H. Wang, A. L. Wyllie, C. B. F. Vogels, R. Earnest, S. Lapidus, I. M. Ott, A. J. Moore, M. C. Muenker, J. B. Fournier, M. Campbell, C. D. Odio, A. Casanovas-Massana, I. T. Yale, R. Herbst, A. C. Shaw, R. Medzhitov, W. L. Schulz, N. D. Grubaugh, C. Dela Cruz, S. Farhadian, A. I. Ko, S. B. Omer, A. Iwasaki, Longitudinal analyses reveal immunological misfiring in severe COVID-19. *Nature* **584**, 463-469 (2020).
 25. I. S. Cheon, C. Li, Y. M. Son, N. P. Goplen, Y. Wu, T. Cassmann, Z. Wang, X. Wei, J. Tang, Y. Li, H. Marlow, S. Hughes, L. Hammel, T. M. Cox, E. Goddery, K. Ayasoufi, D. Weiskopf, J. Boonyaratankornkit, H. Dong, H. Li, R. Chakraborty, A. J. Johnson, E. Edell, J. J. Taylor, M. H. Kaplan, A. Sette, B. J. Bartholmai, R. Kern, R. Vassallo, J. Sun, Immune signatures underlying post-acute COVID-19 lung sequelae. *Sci Immunol* **6**, eabk1741 (2021).
 26. C. Phetsouphanh, D. R. Darley, D. B. Wilson, A. Howe, C. M. L. Munier, S. K. Patel, J. A. Juno, L. M. Burrell, S. J. Kent, G. J. Dore, A. D. Kelleher, G. V. Matthews, Immunological dysfunction persists for 8 months following initial mild-to-moderate SARS-CoV-2 infection. *Nat Immunol* **23**, 210-216 (2022).
 27. Y. Su, D. Yuan, D. G. Chen, R. H. Ng, K. Wang, J. Choi, S. Li, S. Hong, R. Zhang, J. Xie, S. A. Kornilov, K. Scherler, A. J. Pavlovitch-Bedzyk, S. Dong, C. Lausted, I. Lee, S. Fallen, C. L. Dai, P. Baloni, B. Smith, V. R. Duvvuri, K. G. Anderson, J. Li, F. Yang, C. J. Duncombe, D. J. McCulloch, C. Rostomily, P. Troisch, J. Zhou, S. Mackay, Q. DeGottardi, D. H. May, R. Taniguchi, R. M. Gittelman, M. Klinger, T. M. Snyder, R. Roper, G. Wojciechowska, K. Murray, R. Edmark, S. Evans, L. Jones, Y. Zhou, L. Rowen, R. Liu, W. Chour, H. A. Algren, W. R. Berrington, J. A. Wallick, R. A. Cochran, M. E. Micikas, I. S.-S. C.-B. Unit, T. Wrin, C. J. Petropoulos, H. R. Cole, T. D. Fischer, W. Wei, D. S. B. Hoon, N. D. Price, N.

- Subramanian, J. A. Hill, J. Hadlock, A. T. Magis, A. Ribas, L. L. Lanier, S. D. Boyd, J. A. Bluestone, H. Chu, L. Hood, R. Gottardo, P. D. Greenberg, M. M. Davis, J. D. Goldman, J. R. Heath, Multiple early factors anticipate post-acute COVID-19 sequelae. *Cell* **185**, 881-895 e820 (2022).
28. K. M. Littlefield, R. O. Watson, J. M. Schneider, C. P. Neff, E. Yamada, M. Zhang, T. B. Campbell, M. T. Falta, S. E. Jolley, A. P. Fontenot, B. E. Palmer, SARS-CoV-2-specific T cells associate with inflammation and reduced lung function in pulmonary post-acute sequelae of SARS-CoV-2. *PLoS Pathog* **18**, e1010359 (2022).
 29. E. S. Cannizzo, G. M. Bellistri, A. Casabianca, C. Tincati, N. Iannotti, A. Barco, C. Orlandi, A. Monforte, G. Marchetti, Immunophenotype and function of CD38-expressing CD4+ and CD8+ T cells in HIV-infected patients undergoing suppressive combination antiretroviral therapy. *J Infect Dis* **211**, 1511-1513 (2015).
 30. O. A. Odumade, J. A. Knight, D. O. Schmeling, D. Masopust, H. H. Balfour, Jr., K. A. Hogquist, Primary Epstein-Barr virus infection does not erode preexisting CD8(+) T cell memory in humans. *J Exp Med* **209**, 471-478 (2012).
 31. Y. D. Mahnke, T. M. Brodie, F. Sallusto, M. Roederer, E. Lugli, The who's who of T-cell differentiation: human memory T-cell subsets. *Eur J Immunol* **43**, 2797-2809 (2013).
 32. H. Karasuyama, K. Miyake, S. Yoshikawa, Y. Yamanishi, Multifaceted roles of basophils in health and disease. *J Allergy Clin Immunol* **142**, 370-380 (2018).
 33. M. Uguccioni, C. R. Mackay, B. Ochensberger, P. Loetscher, S. Rhis, G. J. LaRosa, P. Rao, P. D. Ponath, M. Baggiolini, C. A. Dahinden, High expression of the chemokine receptor CCR3 in human blood basophils. Role in activation by eotaxin, MCP-4, and other chemokines. *J Clin Invest* **100**, 1137-1143 (1997).
 34. A. E. Weant, R. D. Michalek, I. U. Khan, B. C. Holbrook, M. C. Willingham, J. M. Grayson, Apoptosis regulators Bim and Fas function concurrently to control autoimmunity and CD8+ T cell contraction. *Immunity* **28**, 218-230 (2008).
 35. C. A. Dutertre, E. Becht, S. E. Irac, A. Khalilnezhad, V. Narang, S. Khalilnezhad, P. Y. Ng, L. van den Hoogen, J. Y. Leong, B. Lee, M. Chevrier, X. M. Zhang, P. J. A. Yong, G. Koh, J. Lum, S. W. Howland, E. Mok, J. Chen, A. Larbi, H. K. K. Tan, T. K. H. Lim, P. Karagianni, A. G. Tzioufas, B. Malleret, J. Brody, S. Albani, J. van Roon, T. Radstake, E. W. Newell, F. Ginhoux, Single-Cell Analysis of Human Mononuclear Phagocytes Reveals Subset-Defining Markers and Identifies Circulating Inflammatory Dendritic Cells. *Immunity* **51**, 573-589 e578 (2019).
 36. D. P. Dyer, L. Medina-Ruiz, R. Bartolini, F. Schuette, C. E. Hughes, K. Pallas, F. Vidler, M. K. L. Macleod, C. J. Kelly, K. M. Lee, C. A. H. Hansell, G. J. Graham, Chemokine Receptor Redundancy and Specificity Are Context Dependent. *Immunity* **50**, 378-389 e375 (2019).
 37. C. Gerlach, E. A. Moseman, S. M. Loughhead, D. Alvarez, A. J. Zwijnenburg, L. Waanders, R. Garg, J. C. de la Torre, U. H. von Andrian, The Chemokine Receptor CX3CR1 Defines Three Antigen-Experienced CD8 T Cell Subsets with Distinct Roles in Immune Surveillance and Homeostasis. *Immunity* **45**, 1270-1284 (2016).
 38. N. J. Maurice, M. J. McElrath, E. Andersen-Nissen, N. Frahm, M. Prlic, CXCR3 enables recruitment and site-specific bystander activation of memory CD8(+) T cells. *Nat Commun* **10**, 4987 (2019).

39. Z. Mikhak, J. P. Strassner, A. D. Luster, Lung dendritic cells imprint T cell lung homing and promote lung immunity through the chemokine receptor CCR4. *J Exp Med* **210**, 1855-1869 (2013).
40. T. Imai, K. Hieshima, C. Haskell, M. Baba, M. Nagira, M. Nishimura, M. Kakizaki, S. Takagi, H. Nomiyama, T. J. Schall, O. Yoshie, Identification and molecular characterization of fractalkine receptor CX3CR1, which mediates both leukocyte migration and adhesion. *Cell* **91**, 521-530 (1997).
41. A. Ali, L. M. Canaday, H. A. Feldman, H. Cevik, M. T. Moran, S. Rajaram, N. Lakes, J. A. Tuazon, H. Seelamneni, D. Krishnamurthy, E. Blass, D. H. Barouch, S. N. Waggoner, Natural killer cell immunosuppressive function requires CXCR3-dependent redistribution within lymphoid tissues. *J Clin Invest* **131**, (2021).
42. L. E. Carlin, E. A. Hemann, Z. R. Zacharias, J. W. Heusel, K. L. Legge, Natural Killer Cell Recruitment to the Lung During Influenza A Virus Infection Is Dependent on CXCR3, CCR5, and Virus Exposure Dose. *Front Immunol* **9**, 781 (2018).
43. C. Mionnet, V. Buatois, A. Kanda, V. Milcent, S. Fleury, D. Lair, M. Langelot, Y. Lacoeyille, E. Hessel, R. Coffman, A. Magnan, D. Dombrowicz, N. Glaichenhaus, V. Julia, CX3CR1 is required for airway inflammation by promoting T helper cell survival and maintenance in inflamed lung. *Nat Med* **16**, 1305-1312 (2010).
44. W. Peters, H. M. Scott, H. F. Chambers, J. L. Flynn, I. F. Charo, J. D. Ernst, Chemokine receptor 2 serves an early and essential role in resistance to Mycobacterium tuberculosis. *Proc Natl Acad Sci U S A* **98**, 7958-7963 (2001).
45. E. S. Baekkevold, M. A. Wurbel, P. Kivisakk, C. M. Wain, C. A. Power, G. Haraldsen, J. J. Campbell, A role for CCR4 in development of mature circulating cutaneous T helper memory cell populations. *J Exp Med* **201**, 1045-1051 (2005).
46. P. Schaerli, L. Ebert, K. Willmann, A. Blaser, R. S. Roos, P. Loetscher, B. Moser, A skin-selective homing mechanism for human immune surveillance T cells. *J Exp Med* **199**, 1265-1275 (2004).
47. J. E. Kohlmeier, S. C. Miller, J. Smith, B. Lu, C. Gerard, T. Cookenham, A. D. Roberts, D. L. Woodland, The chemokine receptor CCR5 plays a key role in the early memory CD8+ T cell response to respiratory virus infections. *Immunity* **29**, 101-113 (2008).
48. A. S. Woodward Davis, H. N. Roozen, M. J. Dufort, H. A. DeBerg, M. A. Delaney, F. Mair, J. R. Erickson, C. K. Slichter, J. D. Berkson, A. M. Klock, M. Mack, Y. Lwo, A. Ko, R. M. Brand, I. McGowan, P. S. Linsley, D. R. Dixon, M. Prlic, The human tissue-resident CCR5(+) T cell compartment maintains protective and functional properties during inflammation. *Sci Transl Med* **11**, (2019).
49. A. N. Wein, S. R. McMaster, S. Takamura, P. R. Dunbar, E. K. Cartwright, S. L. Hayward, D. T. McManus, T. Shimaoka, S. Ueha, T. Tsukui, T. Masumoto, M. Kurachi, K. Matsushima, J. E. Kohlmeier, CXCR6 regulates localization of tissue-resident memory CD8 T cells to the airways. *J Exp Med* **216**, 2748-2762 (2019).
50. H. Stenstad, M. Svensson, H. Cucak, K. Kotarsky, W. W. Agace, Differential homing mechanisms regulate regionalized effector CD8 α beta+ T cell accumulation within the small intestine. *Proc Natl Acad Sci U S A* **104**, 10122-10127 (2007).
51. D. Brownlie, I. Rødahl, R. Varnaite, H. Asgeirsson, H. Glans, S. Falck-Jones, S. Vangeti, M. Buggert, H.-G. Ljunggren, J. Michaëlsson, S. Gredmark-Russ, A. Smed-Sörensen, N.

- Marquardt, Distinct lung-homing receptor expression and activation profiles on NK cell and T cell subsets in COVID-19 and influenza. *bioRxiv*, (2021).
52. M. Schorer, N. Rakebrandt, K. Lambert, A. Hunziker, K. Pallmer, A. Oxenius, A. Kipar, S. Stertz, N. Joller, TIGIT limits immune pathology during viral infections. *Nat Commun* **11**, 1288 (2020).
 53. N. M. Provine, P. Klenerman, MAIT Cells in Health and Disease. *Annu Rev Immunol* **38**, 203-228 (2020).
 54. H. Flament, M. Rouland, L. Beaudoin, A. Toubal, L. Bertrand, S. Lebourgeois, C. Rousseau, P. Soulard, Z. Gouda, L. Cagninacci, A. C. Monteiro, M. Hurtado-Nedelec, S. Luce, K. Bailly, M. Andrieu, B. Saintpierre, F. Letourneur, Y. Jouan, M. Si-Tahar, T. Baranek, C. Paget, C. Boitard, A. Vallet-Pichard, J. F. Gautier, N. Ajzenberg, B. Terrier, F. Pene, J. Ghosn, X. Lescure, Y. Yazdanpanah, B. Visseaux, D. Descamps, J. F. Timsit, R. C. Monteiro, A. Lehuen, Outcome of SARS-CoV-2 infection is linked to MAIT cell activation and cytotoxicity. *Nat Immunol* **22**, 322-335 (2021).
 55. J. C. Ribot, N. Lopes, B. Silva-Santos, gammadelta T cells in tissue physiology and surveillance. *Nat Rev Immunol* **21**, 221-232 (2021).
 56. M. S. Davey, C. R. Willcox, S. Hunter, S. A. Kasatskaya, E. B. M. Remmerswaal, M. Salim, F. Mohammed, F. J. Bemelman, D. M. Chudakov, Y. H. Oo, B. E. Willcox, The human Vdelta2(+) T-cell compartment comprises distinct innate-like Vgamma9(+) and adaptive Vgamma9(-) subsets. *Nat Commun* **9**, 1760 (2018).
 57. C. Maucourant, I. Filipovic, A. Ponzetta, S. Aleman, M. Cornillet, L. Hertwig, B. Strunz, A. Lentini, B. Reinius, D. Brownlie, A. Cuapio, E. H. Ask, R. M. Hull, A. Haroun-Izquierdo, M. Schaffer, J. Klingstrom, E. Folkesson, M. Buggert, J. K. Sandberg, L. I. Eriksson, O. Rooyackers, H. G. Ljunggren, K. J. Malmberg, J. Michaelsson, N. Marquardt, Q. Hammer, K. Stralin, N. K. Bjorkstrom, C.-S. G. Karolinska, Natural killer cell immunotypes related to COVID-19 disease severity. *Sci Immunol* **5**, (2020).

REVIEWERS' COMMENTS

Reviewer #1 (Remarks to the Author):

The authors have extensively addressed all comments made by this reviewer. This has improved the manuscript further. Especially the comment on the post-recovery samples which the authors took as representatives for pre-COVID-19 samples/immune profile was challenging but the authors addressed it thoroughly.

I have no additional comments.

Reviewer #2 (Remarks to the Author):

The authors adequately responded to my concerns except for my comment that the title should be modified to reflect that the study findings are ASSOCIATED with severe COVID19 rather than PREDICT COVID-19 severity.

Reviewer #3 (Remarks to the Author):

The authors have addressed most of the comments and have made significant improvement to the clarity of the data presentation and quality of the manuscript.

The main and the most important question that was raised in fact by all three reviewers is the claim that the immune phenotypes and signatures observed in those who have recovered from COVID19 were same/similar to the pre-infection state and were associated severe COVID19. While this has been carefully considered and factored in in the analyses and discussion, it is perhaps inappropriate to make such claim without proper pre-infection controls from the same cohort. Granted that these samples were just not possible to obtain, it is my opinion that the authors must explicitly state this limitation very clearly throughout manuscript but were not explicitly described in the results and discussion.

Point-by-point response to second review

We thank the reviewers for their positive feedback on our revisions. Here, we address their remaining concerns point-by-point. In black is the reviewer comments. Our response is in red font and existing text in grey font, while changes in the text are in blue font.

Reviewer #1 (Remarks to the Author):

The authors have extensively addressed all comments made by this reviewer. This has improved the manuscript further. Especially the comment on the post-recovery samples which the authors took as representatives for pre-COVID-19 samples/immune profile was challenging but the authors addressed it thoroughly.

I have no additional comments.

We agree with the reviewer that the reviewer's comments and revisions improved the manuscript and would like to thank the reviewer for the helpful comments.

Reviewer #2 (Remarks to the Author):

The authors adequately responded to my concerns except for my comment that the title should be modified to reflect that the study findings are ASSOCIATED with severe COVID19 rather than PREDICT COVID-19 severity.

We agree that our approach is challenging and potentially poses the risk that some long-term perturbed immune traits after recovery can be misleadingly defined as baseline immune difference between individuals who develop mild and severe COVID-19. Note the Reviewer's proposed title change makes it sound like we are measuring traits *during* severe COVID, not *before*.. Therefore, we propose to change the title to:

"Baseline Immune phenotypes that determine subsequent COVID-19 severity"

Reviewer #3 (Remarks to the Author):

The authors have addressed most of the comments and have made significant improvement to the clarity of the data presentation and quality of the manuscript.

The main and the most important question that was raised in fact by all three reviewers is the claim that the immune phenotypes and signatures observed in those who have recovered from COVID19 were same/similar to the pre-infection state and were associated severe COVID19. While this has been carefully considered and factored in in the analyses and discussion, it is perhaps inappropriate to make such claim without proper pre-infection controls from the same cohort. Granted that these samples were just not possible to obtain, it is my opinion that the authors must explicitly state this limitation very clearly throughout manuscript but were not explicitly described in the results and discussion.

We thank the reviewer for the comment. We agree that our approach is unconventional and that we need to clearly describe our approach and highlight its limitations. We mention this in the results section, Page 6, line 141:

We aimed to identify immune signatures at steady-state which contribute to severe COVID-19. However, cohorts with baseline PBMC samples from patients who had not yet been infected with COVID-19 are not available. In this cross-sectional study, we analyzed PBMC collected after recovery from mild, moderate, severe and critical COVID-19 (Extended Data Figs. 1 and 2) and focused on traits related to the highly significant results from GWAS analysis.

We continue to address the caveat that permanently perturbed immune traits with no recovery within the first 100 days post infection could be misleadingly defined as baseline and result in a false hit in our analysis. However, given the immune system's capabilities to return to pre-infection baseline in various diseases makes such a scenario unlikely. This is stated in the results section, Page 7, line 155:

COVID-19 induced immune perturbations can persist after viral clearance and recovery (37, 38). We hypothesized that immune cells may show three potential trajectories following infection. First, a cell trait may not be affected by COVID-19 infection and remain at baseline throughout infection (but still contribute to susceptibility to severe disease). Secondly, a trait may be affected and deviate from unexposed healthy individuals only during active COVID-19 and recover over time after pathogen clearance. Lastly, a trait may change rapidly after viral transmission and remain perturbed after viral clearance without improvement. The latter is likely infrequent, if at all, due to the renewal capacity of the human immune system (39-44). The second scenario would result in temporal changes of immune traits which would be detectable in samples collected early vs late after symptom onset. To identify traits that might contribute to COVID-19 severity, we took the conservative approach of eliminating traits that showed any evidence of perturbation over time, for which we cannot assert the baseline, pre-COVID-19 value. A caveat to this approach is that a trait which changed from baseline before our earliest measurements, and then remained at that level following our latest measurements, would be included in our analyses. However, we considered that scenario highly unlikely in this dynamic acute infection, particular since viral clearance often occurs early irrespective of disease severity (37) and given the immune system's capacity to recover rapidly from perturbations (39-44).

Furthermore, we discuss extensively the literature about immune restoration after recovery from COVID-19 and several other viral infections in the Discussion section, Page 15, line 464:

Similarly, most SARS-CoV2-induced immune perturbations normalized within 60-80 days after symptom onset (37, 38, 75). Furthermore, immune perturbations in nasal tissues from severe COVID-19 patients improved already within 15-61 days of hospitalization and normalized after discharge (median of 77 days after symptom onset). Bergamaschi et al. demonstrated that viral titers drop to undetected levels irrespective of disease severity between 13-24 days after symptom onset which overlapped with improvement of immune perturbations (37). Nevertheless, some immune perturbations lasted for months and included naïve B cells, activated CD38+HLA-DR+ T cells and the expression of CXCR3 (37, 38, 75). Noteworthy, not all the long-term immune perturbations described by Bergamaschi et al. could be replicated in studies with a more detailed focus on individual immune cell subsets including pDCs (76) and MAIT cells (77).

We added text to the discussion section (Page 15, line 448) stating that pre-infection samples would be the most precise approach to determine pre-infection immune traits which are predictive of COVID-19 severity. However, since such samples are impossible to obtain, our approach is a reasonable alternative:

Samples collected prior to infection are most suitable to study pre-infection immune signatures. However, pre-infection samples are difficult to obtain for acute emerging diseases such as COVID-19. Thus, we measured the immune composition using high-dimensional flow cytometry in peripheral blood of individuals recovered from mild, moderate, severe and critical COVID-19 to identify immune signatures associated with COVID-19 severity. The human immune system rapidly reverts to baseline after pathogen clearance with a composition comparable to pre-infection (65, 66). Therefore, probing the immune system after disease recovery serves as an alternative to identify baseline immune signatures which determine disease severity.

Our approach has the caveat that rapidly and permanently perturbed immune signatures will be identified as baseline. However, rapid immune recovery has been described for several acute viral infections including Ebola (67), Zika (68), Dengue (69, 70), Measles (71), Hantavirus (72) and Respiratory Syncytial Virus (73) as well as live virus vaccines for Smallpox and Yellow Fever (74). In these cases, most immune perturbations recovered within weeks after pathogen clearance. Therefore, it is unlikely that immune perturbations occur permanently without any sign of improvement over time. Such traits would, in any case, represent a small fraction of the traits we observed.